# Replication competent HIV-guided CRISPR screen identifies antiviral factors including targets of the accessory protein Nef

Caterina Prelli Bozzo [1,3], Alexandre Laliberté [1,3], Aurora De Luna [1], Chiara Pastorio [1], Kerstin Regensburger[1], Stefan Krebs [2], Alexander Graf [2], Helmut Blum [2], Meta Volcic [1], Konstantin M. J. Sparrer [1] ✉ & Frank Kirchhoff [1] ✉

Innate antiviral factors are essential for effective defense against viral pathogens. However, the identity of major restriction mechanisms remains elusive. Current approaches to discover antiviral factors usually focus on the initial steps of viral replication and are limited to a single round of infection. Here, we engineered libraries of >1500 replication-competent HIV-1 constructs each expressing a single gRNAs to target >500 cellular genes for virus-driven discovery of antiviral factors. Passaging in CD4+ T cells robustly enriched HIV-1 encoding sgRNAs against *GRN*, *CIITA*, *EHMT2*, *CEACAM3*, *CC2D1B* and *RHOA* by >50-fold. Using an HIV-1 library lacking the accessory *nef* gene, we identified IFI16 as a Nef target. Functional analyses in cell lines and primary CD4+ T cells support that the HIV-driven CRISPR screen identified restriction factors targeting virus entry, transcription, release and infectivity. Our HIV-guided CRISPR technique enables sensitive discovery of physiologically relevant cellular defense factors throughout the entire viral replication cycle.

Viral pathogens and their hosts are caught in an ever-ongoing arms race. Cellular antiviral factors are an essential part of the innate immune system and may provide immediate and broad protection against viral pathogens[1–3]. However, viruses adapt and have evolved sophisticated mechanisms to evade or counteract antiviral defense mechanisms[4–6]. Thus, the outcome of viral exposures depends on complex pathogen-host interactions and failure of cellular defense mechanisms may result in severe disease and - in the worst case - devastating pandemics.

HIV-1 is the causative agent of the AIDS pandemic and a serious challenge to public health for over 40 years. Studies of HIV-1 and related lentiviruses allowed the discovery of a complex repertoire of restriction factors (RFs) that have the potential to inhibit viral pathogens at essentially every step of their replication cycle[1–3]. They further revealed that differences in the ability to counteract antiviral RFs

explain why only one of at least thirteen independent zoonotic lentiviral transmissions resulted in the AIDS pandemic[6–8]. However, we are only beginning to understand the complex interplay between HIV and its human host and important antiviral factors remain to be discovered. For example, the determinants of interferon (IFN) resistance of transmitted/founder (TF) HIV-1 strains that are responsible for primary infection[9,10] frequently map to regions in the viral genome unlikely to affect the susceptibility to known restriction factors. Previous studies further suggest that targets of the HIV-1 accessory Vif, Vpu, Vpr and Nef proteins which counteract major antiviral factors remain to be identified[11,12].

The discovery of yet unknown antiretroviral factors is of broad interest and relevance as RFs not only restrict HIV-1 but a wide range of viral pathogens and frequently also play roles in inflammation and cancers[13–21]. Thus, new insights into antiviral defense mechanisms will

[1]Institute of Molecular Virology, Ulm University Medical Center, 89081 Ulm, Germany. [2]Laboratory for Functional Genome Analysis Gene Center, LMU Munich, 81377 Munich, Germany. [3]These authors contributed equally: Caterina Prelli Bozzo, Alexandre Laliberté. ✉e-mail: Konstantin.Sparrer@uni-ulm.de; Frank.Kirchhoff@uni-ulm.de

not only improve our ability to combat viral pathogens but also help to develop innovative therapies against other diseases. However, discovery of restriction factors is a challenging task since they are structurally and functionally highly diverse and there are no generally applicable criteria for their identification[2,4]. Previous studies used expression libraries of IFN-stimulated genes (ISGs), RNA interference (RNAi), and pooled CRISPR/Cas9 screens[22]. However, over-expression screens can usually not be performed genome-wide and are prone to artifacts[22–24]. RNAi screens show low reproducibility and high rates of false positives due to inefficient knock-down and off-target effects[22,25]. Most CRISPR/Cas9-based screens involve the introduction of pooled sgRNAs into Cas9-expressing cells via lentiviral transduction[22] already altering the innate immune landscape of the cells prior to virus infection. Subsequently, cells showing resistance or increased sensitivity in single-round virus infection assays are enriched to identify pro- or antiviral factors, respectively. So far just a single targeted CRISPR-based screen for HIV restriction factors has been reported[26]. In this approach, THP-1 cells are first transduced with LTR-containing lentiviral constructs expressing both Cas9 and sgRNAs to knock-out potential antiviral genes. On the next day, the cells are treated with IFN-α and infected with wild-type HIV-1. Three days later, the cultures are examined for enrichment of specific sgRNAs in HIV-1 virions compared to their frequency in genomic DNA by RT-PCR/PCR and deep sequencing[26]. The initial screen identified IFN-induced antiviral factors[26] and has subsequently been modified to identify cellular that promote HIV-1 replication[27], restrict HIV-1 by targeting the viral capsid[28], or affect viral latency[29]. In addition, it has been used to identify ISGs restricting HIV-1 in primary CD4+ T cells[30]. While this CRISPR-knockout screen allowed significant discoveries, it requires transduction with lentiviral Cas9/gRNA expression vectors prior to infection with wild-type HIV-1 and specifically detects cellular factors affecting co-packing of the lentiviral Cas9/gRNA encoding RNA into HIV-1 particles after a single round of virus replication.

Previous genetic screens provided some insights into virus-host interactions. However, they have significant limitations. Most of them rely on manipulation of the cells prior to virus infection. Thus, they will miss factors important for cell survival and may yield misleading results because some cellular proteins have different effects in uninfected and infected cells. For example, viral receptors are essential for viral entry but usually impair virus release and the infectivity of progeny virions. Importantly, current overexpression, RNA interference, and CRISPR/Cas9 screens usually involve only single-cycle infections. They frequently detect only factors affecting early steps of the viral replication cycle and are usually not very sensitive. Since cells exert numerous defense mechanisms the contribution of individual factors to the control of viral pathogens may seem small. However, a 2-fold growth advantage may result in >1000-fold higher virus yields after just 10 rounds of replication. Thus, effects that may be missed in single round of infection screens can have a major impact on viral spread in vivo. Altogether, robust, sensitive, broad and versatile screens that unravel physiologically relevant antiviral factors and mechanisms are urgently needed for a better understanding of the complex host-pathogen interplay.

To address this, we combined the CRISPR/Cas9 technology with the selection power of replication-competent HIV-1. Specifically, we equipped full-length infectious molecular clones (IMCs) of HIV-1 with molecular tools (i.e. sgRNAs) allowing the virus to eliminate antiviral genes but at the same time revealing their identity. We named this technique the traitor virus (TV) approach since the pathogen itself identifies its cellular opponents. Our results demonstrate that TVs targeting specific antiviral genes show increased replication fitness in Cas9 expressing T cells and are rapidly enriched in emerging viral populations. We obtained initial insights into the underlying mechanisms and confirmed the antiviral effect of several factors in primary human CD4+ T cells. Finally, utilization of "handicapped" *nef*-defective TVs allowed the discovery of IFI16 as a target of the viral accessory protein Nef. In summary, we show that the TV-guided technology allows the robust and effective identification of antiviral cellular factors including not yet appreciated targets of the HIV-1 accessory proteins providing in-depth insights into virus-host interactions.

## Results

### Design and proof-of-principle of the TV approach

To allow efficient virus-driven discovery of antiviral cellular genes, we generated replication-competent HIV-1 constructs encoding single guide RNAs (sgRNAs). Our goal was to equip HIV-1 with sequence and target-specific genetic "scissors" to generate "traitor viruses" (TVs) whose replication fitness reveals their cellular opponents. To achieve this, we inserted a cassette encompassing the human U6 promoter, sgRNAs comprised of the flexible targeting region, and an invariant scaffold into the proviral genome of the well-characterized HIV-1 NL4-3 infectious molecular clone (IMC) (Fig. 1a). The resulting proviral HIV-1 constructs express all viral genes under the control of the LTR promoter and via the regular splice sites. However, proviral integration into the host genome initiates U6-driven expression of sgRNAs and editing of their target genes in the presence of Cas9. Thus, similar to having additional accessory genes, the engineered viruses themselves drive the countermeasures against cellular defense mechanisms. The sgRNA expression cassette encompasses only ~351 nucleotides and hence increased the size of the viral RNA genome that is packaged into HIV-1 particles only moderately from 9833 to 10,184 base-pairs. We hypothesized that this insertion should be well tolerated and enhance the replication fitness of TV variants expressing sgRNAs inactivating cellular genes that suppress steps in the viral replication cycle after proviral integration or reduce the infectiousness of progeny HIV-1 particles.

To determine whether this approach works in principle, we engineered TV constructs encoding non-targeting (NT) control sgRNAs and four unique sgRNAs targeting two established restriction factors: tetherin that inhibits virion release[31,32] and GBP5, which impairs viral infectivity by suppressing furin-mediated processing of envelope glycoproteins (Fig. 1b)[33,34]. For passaging, we generated CEM-M7 cells stably expressing Cas9. This T/B hybrid cell line expresses CD4, CCR5 and CXCR4 and contains the *GFP* reporter gene under the control of the HIV-1 LTR[35]. Since many antiviral factors including tetherin and GBP5 are IFN-inducible, infections were performed in the absence and presence of IFN-β. Viral supernatants were collected at different days post-infection and the abundance of sgRNAs in the viral genomes determined by qRT-PCR. We observed efficient viral replication and enrichment of TVs expressing sgRNAs targeting genes encoding the restriction factors (Fig. 1b). FACS analyses confirmed that the selected HIV-1 U6-sgRNA-scaffold constructs reduced tetherin and GBP5 expression by 70% and 40%, respectively (Fig. 1c, Supplementary Fig. 1a). Altogether, these results provided proof-of-concept that replication-competent HIV-1 TV constructs allow effective selection of sgRNAs targeting antiviral genes.

### Generation and optimization of HIV-1 U6-sgRNA-scaffold libraries

To identify antiviral restriction factors, we generated a library of TV constructs targeting genes encoding 511 different candidate restriction factors (CRFs; Supplementary Data 1) each by three unique sgRNAs. A total of 200 target genes were chosen based on previous analyses of 15,052 protein-coding genes that revealed shared features of known antiviral factors, such as the in vivo response to HIV-1 infection and/or IFNs, codon-specific positive selection, burden of synonymous, missense and non-sense variation, as well as the number of paralogs[24]. The remaining factors were selected because of their putative roles in pathogen sensing or in the various steps of the HIV-1 replication

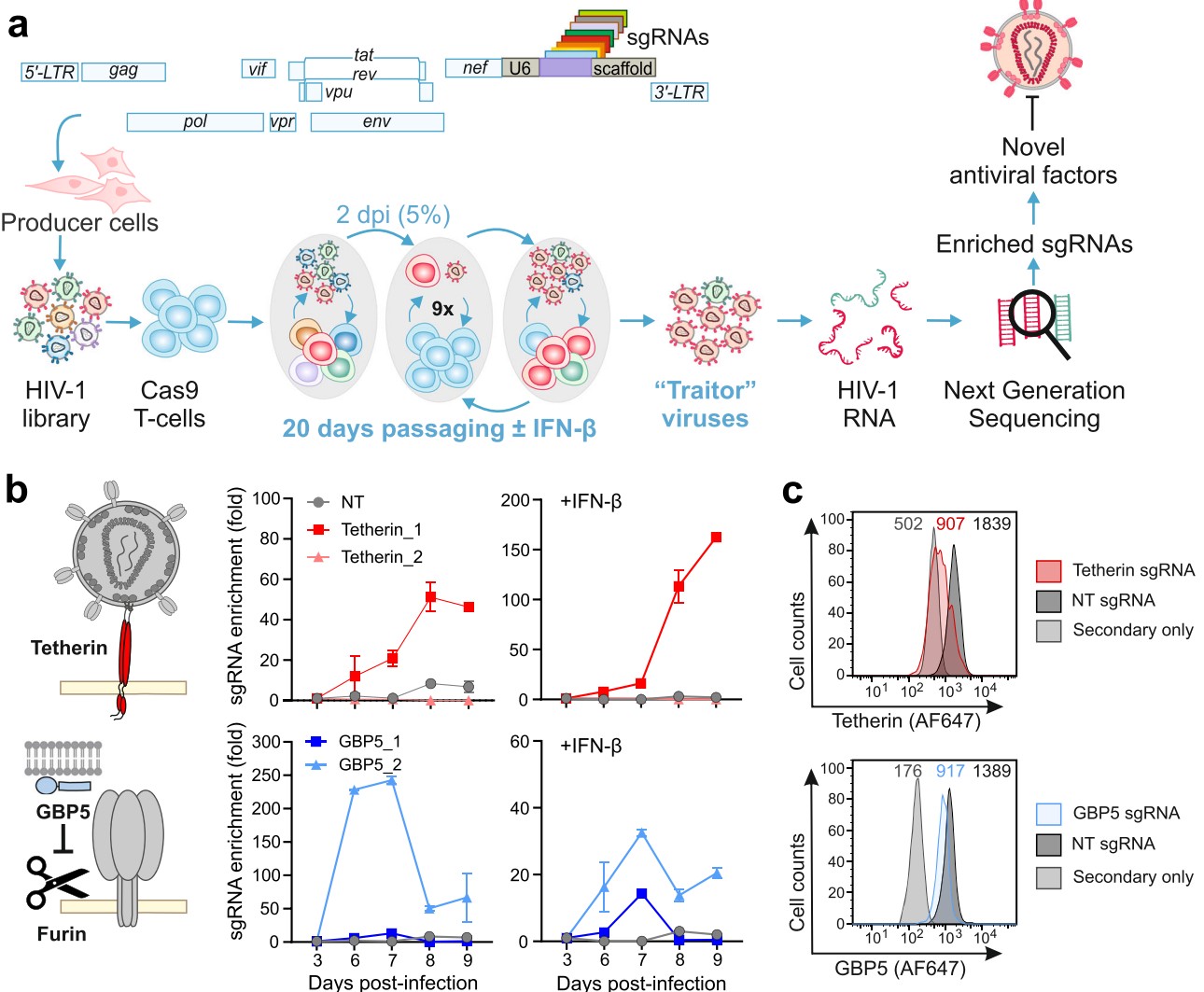

**Fig. 1 | Assay principle and proof of concept. a** Outline of the CRISPR/Cas9-based virus-guided discovery approach. Proviral HIV-1 constructs are engineered to contain the sgRNAs expression cassette between the *nef* gene and the 3′LTR. To produce virus stocks, HEK293T cells are transfected with libraries of HIV-1 constructs expressing various sgRNAs. The resulting swarms of HIV-1 sgRNA viruses are passaged every two days in Cas9-expressing cells in the presence or absence of IFN-β. Viral supernatants are harvested every five days and the frequencies of HIV-1 sgRNAs are determined by next-generation sequencing. Target genes of sgRNAs that are selected and hence are associated with an advantage for viral replication are cloned and examined for their antiviral activity and mechanism. Note that the U6-sgRNA-scaffold region is not to scale. **b** Enrichment of HIV-1 NL4-3 expressing sgRNAs targeting *tetherin* (red), *GBP5* (blue) or non-targeting (NT, gray). The left panel provides a schematic showing tetherin trapping HIV-1 particles at the cell surface and inhibition of furin-mediated processing of the gp160 Env precursor to mature gp120 and gp41. The right panels show the enrichment of the indicated sgRNAs at different days post-infection in presence and absence of IFN-β. Relative enrichment was quantified using SYBR green qRT-PCR. Dots represent the mean of $n = 3 \pm$ SEM. **c** Flow cytometry analysis of tetherin and GBP5 expression levels in CEM-M7-Cas9 infected with HIV-1 NL4-3 expressing either Tetherin_1 (red), GBP5_2 (blue) or NT (gray) sgRNA as indicated.

cycle[36]. As controls, we used eleven non-targeting sgRNAs. The sequences of the sgRNAs were selected from the GeCKO V2 library[37] with the lowest off-target scores and targeting various exons of their respective target gene. Cloning into the proviral HIV-1 NL4-3 constructs was highly efficient and measurements of colony forming units indicated an average coverage of ~1.000 individual transformants per sgRNA. Transfection of the proviral TV-NL4-3-CRF-sgRNA library yielded high levels of infectious HIV-1 (TCID$_{50}$ of $7.34 \times 10^6$ per ml virus stock) that replicated efficiently in CEM-M7-Cas9 cells. However, the quality of sequence reads rapidly declined and PCR analyses confirmed loss of the U6-sgRNA-scaffold cassette in most replicating HIV-1 variants during passage (Supplementary Fig. 1b). Sequence analyses revealed that deletions were mediated by recombination between repeats flanking the U6-sgRNA-scaffold sequence (Supplementary Fig. 1b, c). Specifically, the accessory *nef* gene overlaps the U3 region of the

3′ LTR and contains *cis*-acting elements, i.e. a T-rich region, polypurine tract (PPT) and attachment (*att*) sequences, required for reverse transcription and integration. The initial TV constructs contain these sequences, referred to as TPI-region hereafter, in both *nef* and at the beginning of the 3′LTR. To remove repetitive hotspots for recombination, we introduced 16 synonymous nucleotide changes in the *nef* open reading frame (Supplementary Fig. 1c). In addition, we mutated the 3′end of the *nef* gene representing a second less prominent site for recombination. The optimized TV-NL4-3-CRF-sgRNA constructs were replication-competent and highly stable during cell-culture passaging (Supplementary Fig. 1d). Thus, we introduced the 1537 different sgRNAs into the optimized backbone using homologous recombination. The proviral TV libraries yielded high levels of infectious virus after transfection into HEK293T cells (Supplementary Fig. 1e) and the mutations in the *nef* coding region did not compromise Nef expression

(Supplementary Fig. 1f). Transformation of the proviral DNA library into *E. coli* resulted in ~$3 \times 10^5$ colonies/ml suggesting sufficient coverage to retain complexity. Deep sequencing confirmed that all 1537 sgRNAs were efficiently cloned into the backbone and that the genomic HIV-1 RNA sequences in the viral stocks reflected those in the proviral DNA TV library (Supplementary Fig. 1g). Altogether, our results showed that silent mutations in the TPI-region of *nef*, together with intact *cis*-regulatory elements downstream of the U6-sgRNA-scaffold cassette and upstream of the core enhancer in the 3′LTR, allow efficient HIV-1 replication and stable sgRNA expression.

## TVs reveal cellular factors restricting HIV-1 replication

To identify sgRNAs associated with a fitness advantage for HIV-1 replication, we infected CEM-M7-Cas9 cells with the pool of infectious TV-NL4-3-CRF-sgRNA viruses targeting 511 potential antiviral genes. CXCR4-tropic HIV-1 NL4-3 IMCs replicate with fast kinetics. For passaging, we thus inoculated uninfected cells with 5% (v/v) of cell cultures obtained 2 days post-infection over a total period of 20 days (Fig. 1a). Virus containing culture supernatants were isolated in 5-day intervals. TV-NL4-3-CRF-sgRNA viruses spread efficiently and produced high levels of infectious virus (Supplementary Fig. 2a). Next generation sequencing (NGS) followed by bioinformatic analysis using MAGeCK[38] revealed the selection of TVs expressing sgRNAs targeting specific candidate antiviral genes (Fig. 2a). Widening volcano plots illustrate that viruses containing sgRNAs conferring a replicative advantage are increasing over time (Fig. 2b). TVs expressing sgRNAs targeting genes encoding Progranulin (GRN), Class II MHC transactivator (CIITA), Coiled-Coil and C2 Domain Containing 1B (CC2D1B), Carcinoembryonic antigen-related cell adhesion molecule 3 (CEACAM3), Heme Oxygenase-1 (HMOX1) and Euchromatic Histone Lysine Methyl-transferase 2 (EHMT2, also named G9a) were increasingly enriched by up to several orders of magnitude (Fig. 2c, Supplementary Fig. 2b). The efficiency of selection varied between different sgRNAs targeting the same gene. However, the impact of individual sgRNAs on viral fitness was confirmed in the presence of IFN-β (Fig. 2c) and highly reproducible in independent experiments (Fig. 2d). TVs expressing sgRNAs targeting known RFs like *TRIM5*, *IFI16*, *IFITM2*, *SAMHD1*, *GBP5* and *IFITM1* were enriched after 15 and 20 days post-infection confirming that targeting RFs provides a selection advantage to the respective viruses (Fig. 2e). In line with our data in CEM-M7-Cas9 cells, TVs targeting *GRN*, *CIITA*, *CC2D1B*, *CEACAM3*, *HMOX1* and *EHMT2* also showed increased fitness in SupT1-Cas9 cells (Fig. 2f, g, Supplementary Fig. 2c). Altogether, efficient and robust enrichment of the same specific sgRNAs in different experimental settings clearly indicated targeting of cellular genes suppressing HIV-1 replication.

## GRN, CIITA and CEACAM3 restrict HIV-1 replication in primary CD4⁺ T cells

To assess the significance of factors identified by the TV approach, we first confirmed that the protein products of genes targeted by sgRNAs associated with increased fitness, i.e. GRN, CIITA, CC2D1B, CEACAM3, HMOX1 and EHMT2 are expressed in the cell lines used for selection (Fig. 3a). In agreement with our finding that TVs targeting the respective genes are selected in the presence and absence of IFN-β (Fig. 2), these six factors were expressed but (unlike ISG15 or tetherin) not further induced by IFN treatment (Fig. 3a). For functional analyses, we initially focused on GRN as sgRNAs targeting the corresponding gene provided a substantial and robust fitness advantage (Fig. 2). GRN expresses an 88 kDa precursor, progranulin (GRN) that has been reported to suppress HIV-1 transcription by interacting with cyclin T1[39,40]. In line with this, partial knock-out (KO) of GRN in CEM-M7-Cas9 cells significantly increased HIV-1 infection (Fig. 3b, Supplementary Fig. 3a). Overexpression of GRN slightly reduced infectious virus production and protein expression of NL4-3 and (more clearly) the HIV-1 CH077 transmitted-founder IMC[9] in transfected HEK293T cells.

Defects in viral accessory genes had little impact on the susceptibility of HIV-1 to GRN (Fig. 3c). In support of an effect on viral transcription, GRN inhibited LTR-driven luciferase expression in the absence and presence of Tat (Fig. 3d). In addition, GRN reduced GFP expression by three proviral HIV-1 IRES-eGFP constructs in a dose-dependent manner (Fig. 3e, Supplementary Fig. 3b). To further examine the significance of the antiviral activity of GRN, we established a sgRNA/Cas9-based KO approach for specific genes in primary CD4⁺ T cells (Supplementary Fig. 3c). KO of GRN reduced its protein levels by ~70% (Fig. 3f) and increased infectious virus production by WT HIV-1 NL4-3 and CH077 IMCs by ~2-fold (Fig. 3g, Supplementary Fig. 3d).

We next examined the effects of CIITA, CC2D1B and CEACAM3. Overexpression of these cellular factors had differential effects. CIITA had no significant impact on infectious virus yields (Fig. 4a) and increased LTR-driven eGFP production by proviral HIV-1 IRES-eGFP IMCs at very high expression levels (Supplementary Fig. 4a). In comparison, CC2D1B inhibited infectious virus production in a dose-dependent manner (Fig. 4a). Western blot analyses showed that CC2D1B significantly reduces virus release and envelope (Env) processing (Supplementary Fig. 4b–d). This agrees with previous data showing that CC2D1A interferes with HIV-1 budding and that both CC2D1A and CC2D1B interact with the CHMP4 subunit of the ESCRT-III complex[41,42]. In contrast, high levels of CEACAM3 over-expression increased infectious virus production by transfected HEK293T cells (Fig. 4a).

To examine effects under more physiological conditions, we performed KO experiments in primary CD4⁺ T cells. Protein expression was reduced by ~60% for CIITA and by ~80% for CC2D1B, while no reduction was observed for CEACAM3 (Supplementary Fig. 4e). Notably, CEACAM3 is part of a large family of closely related adhesion molecules and antibody cross-reactivities may obscure KO effects in western blots. However, further analysis using a more specific flow cytometry antibody revealed a ~90% KO efficiency of CEACAM3 (Supplementary Fig. 4f, g). CIITA is a transcription factor regulating MHC class II promoters[43,44] including HLA-DR. In line with this, KO of CIITA but not GRN reduced HLA-DR expression levels (Supplementary Fig. 4h, i). Treatment with both *CIITA* and *CEACAM3* targeting sgRNAs increased HIV-1 NL4-3 and CH077 replication in primary CD4⁺ T cells by 2- to 3-fold, while KO of CC2D1B had no enhancing effect (Fig. 4b–d, Supplementary Fig. 4j). To further examine the role of CC2D1B, we bypassed the early step of regular HIV-1 infection by utilizing *env*-defective single-round HIV-1 particles pseudo-typed with the VSV-G protein. In agreement with the inhibitory effect of CC2D1B over-expression on virus release (Supplementary Fig. 4b, c), reduced CC2D1B expression moderately increased p24 antigen production under these experimental conditions (Fig. 4e). Altogether, our results showed that 3 of the 4 factors identified by the TV approach (GRN, CIITA and CEACAM3) restrict HIV-1 replication in primary CD4⁺ T cells. The remaining one (CC2D1B) reduced infectious virus release in overexpression assays and in primary CD4⁺ T cells infected with VSV-G-pseudo-typed HIV-1 particles. Notably, KO of these factors had no significant effects on viability and metabolic activity of primary human CD4⁺ T cells (Supplementary Fig. 5a, b). This illustrates the power of TV-based screens in identifying relevant antiviral factors and further shows that some of them would be missed in commonly used over-expression and KO assays.

## Increased fitness of CH077-based TVs targeting *HMOX1*, *EHMT2*, *CC2D1B* and *RHOA*

Initially, we utilized NL4-3 because this HIV-1 IMC has been characterized and proven useful in numerous previous studies. However, NL4-3 is adapted for efficient replication in T cell lines. Thus, cellular factors restricting replication of primary patient-derived HIV-1 strains may be missed. To address this, we generated TV libraries of HIV-1 CH077 representing a TF HIV-1 IMC capable of using both CCR5 and

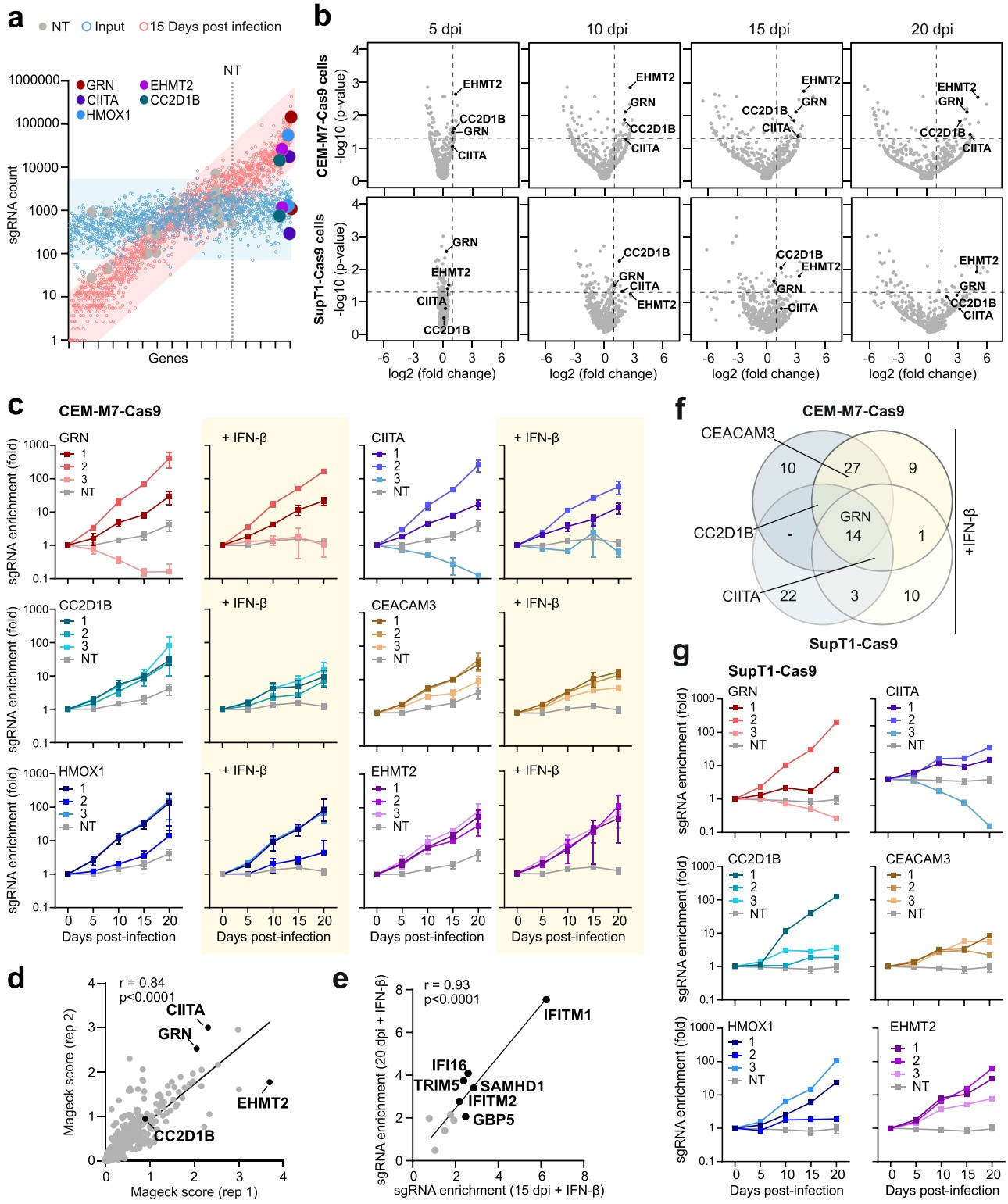

CXCR4 for viral entry[9]. The initial constructs contain a duplication of the TPI and U3 regions in *nef* and the 3'LTR. To minimize recombination, we codon optimized the *nef* gene with changes similar to the stabilizing changes in the NL4-3 proviral genome (Supplementary Figs. 1c, 6a). The optimized CH077 U6-sgRNA-scaffold construct expressed functional Nef (Supplementary Fig. 6b). Viral infectivity was moderately reduced compared to the parental CH077 IMC (Supplementary Fig. 6c), presumably due to the slightly increased size of the viral genome. Nonetheless, the viral titers were well sufficient to cover

all 1537 sgRNAs and NGS confirmed that the virus stocks faithfully represented the proviral CH077-CRF-sgRNA library (Supplementary Fig. 6d). TV-CH077-CRF-sgRNA viruses spread efficiently but with slower kinetics than NL4-3 and produced high levels of infectious virus for ≥30 days of passaging (Supplementary Fig. 6e).

Replication of CH077-based TVs resulted in the selection of an overlapping but distinct set of sgRNAs compared to the NL4-3-based library (Fig. 5a, b). For example, the screen with the primary HIV-1 IMC confirmed that sgRNAs targeting *GRN*, *CIITA*, *CEACAM3*, *HMOX1* and

**Fig. 2 | Selection of HIV-1 U6-CRF-sgRNA constructs targeting *GRN*. a** Scatter plot of individual sgRNA counts in CEM-M7-Cas9 cells supernatants 15 days post infection (red outlines) and in the input (blue outlines) versus gene names sorted by fold enrichment. NT and dotted line indicate the occurrence of the first non-targeting control sgRNA. Selected factors are highlighted by colors as indicated. **b** Volcano plots indicating specific target genes of which the sgRNAs are significantly enriched during passage in CEM-M7 (upper) or SupT1 (lower) Cas9 cells at indicated days post infection (dpi). Dashed lines indicate *p* value 0.05 and 2-fold change on Y and X axis, respectively. *P* value, negative binomial (NB) model with robust ranking aggregation (RRA). **c** Read counts relative to input virus from the MAGeCK analysis showing the enrichment of sgRNAs targeting *GRN* (red), *CIITA* (purple), *CC2D1B* (turquoise), *CEACAM3* (brown), *HMOX1* (blue) or *EHMT2* (violet) in presence or absence of IFN-β in CEM-M7-Cas9 cells over time. NT, gray. Dots represent the mean of $n = 3 \pm$ SEM (independent experiments). **d** Correlation between MAGeCK score obtained in two independent experiments in CEM-M7 cells. Pearson's correlation, r value and *p*-value are indicated. **e** Correlation between the enrichment of known RFs at the 15- and 20-day time-points in presence of IFN-β in CEM-M7-Cas9 cells. Pearson's correlation, r value and *p*-value are indicated. **f** Venn diagram illustrating the sgRNAs that were enriched in the different conditions (e.g. different cell lines or in presence or absence of IFN-β). Genes were considered enriched when the -log$_{10}$ of the positive MAGeCK score was above or equal 1.5. **g** Read counts relative to input virus from the MAGeCK analysis showing the enrichment of sgRNAs targeting *GRN* (red), *CIITA* (purple), *CC2D1B* (turquoise), *CEACAM3* (brown), *HMOX1* (blue) or *EHMT2* (violet) in SupT1-Cas9 cells over the course of 20 days. NT, gray. Dots represent $n = 1$. Error bars of the NT sample represents the mean of 11 NT sgRNAs ±SEM.

*EHMT2* are associated with increased replication fitness (Fig. 5c). Notably, sgRNAs targeting *CC2D1B* were more efficiently selected by CH077-based compared to NL4-3-based TVs and sgRNAs targeting *RHOA* only increased fitness of CH077 but not NL4-3 in presence of IFN-β (Fig. 5d, e). Ras homolog gene family member A (RHOA) is a small GTPase involved in actin cytoskeleton dynamics, cell motility and regulation of innate immunity[45]. Both enhancing and inhibitory effects on HIV-1 have been reported[46,47]. Overexpression of RHOA had no effect on HIV-1 NL4-3 but moderately affected infectious virus production of primary virus strains (Supplementary Fig. 7a). KO of RHOA was highly efficient but reduced, rather than enhanced, HIV-1 replication (Supplementary Fig. 7b, c). Lack of RHOA had only marginal effects on cell viability (Supplementary Figs. 5a, 7d). In addition, it increased metabolic activity (Supplementary Fig. 5b) but impaired cell proliferation (Supplementary Fig. 7e, f). These effects may explain why HIV-1 replication was reduced if RHOA is depleted prior to infection. Altogether, the TV-CH077-based screen confirmed the power of virus-driven identification of antiviral factors and suggests roles of CC2D1B and RHOA in limiting primary HIV-1 replication.

### Nef-defective TVs reveal potential Nef targets
We hypothesized that lack of specific accessory genes will increase the selective advantage mediated by sgRNAs targeting restriction factors that are otherwise counteracted by these viral factors. To address this, we generated a TV-CRF-sgRNA library using an otherwise isogenic *nef*-deleted HIV-1 NL4-3 as a backbone (Fig. 6a). We found that the Δ*nef*-TV-NL4-3-CRF-sgRNA viruses replicated with moderately faster kinetics in CEM-M7-Cas9 cells compared to the parental constructs (Supplementary Fig. 8a, b). It has been shown that lack of intact *nef* genes promotes additional deletions in the *nef*-unique and U3 region of the viral LTR that accelerate viral replication[48,49]. Thus, reduction of the genome size by 360 bp may explain faster replication kinetics of the Δ*nef*-TV-NL4-3-CRF-sgRNA viruses, although other effects, e.g. on viral RNA stability, might also play a role. Altogether, results obtained using WT and Δ*nef* backbones correlated well and confirmed that sgRNAs targeting *GRN*, *HMOX1*, *CIITA* and *EHMT2* increase viral fitness (Fig. 6b, Supplementary Fig. 8c, d). Lack of Nef was associated with moderately increased selection efficiency of sgRNAs targeting *IRF-3* (Supplementary Fig. 8d). IRF-3 is a major transcriptional regulator of type I IFN-dependent immune responses suggesting that they might be more effective against *nef*-deficient HIV-1. Predictably, lack of Nef promoted the selection of TVs expressing sgRNAs targeting genes encoding factors that are known to be counteracted by Nef, such as SERINC5[50,51] (Fig. 6c–e). The abundance of *SERINC5*-targeting sgRNAs increased with relatively slow kinetics (Fig. 6d), possibly because this factor affects virion infectivity and hence the inhibitory effect only becomes apparent over several rounds of replication.

Lack of an intact *nef* gene also increased the efficiency of selection for TV sgRNA variants targeting *IFI16*, especially in the presence of IFN-β (Fig. 6f–h). This came as surprise since we have previously shown that IFI16 inhibits most subtypes of HIV-1 by sequestering the transcription factor Sp1 and that clade C viruses evade this restriction by acquisition of an additional NF-kB binding site[52,53]. Analysis of five pairs of WT and *nef*-defective HIV-1 strains including two primary subtype B and two clade C IMCs confirmed that the latter are less sensitive to the inhibitory effects of IFI16 (Fig. 6i). In all cases, however, an intact *nef* gene clearly reduced viral susceptibility to IFI16 restriction (Fig. 6i). In contrast, expression of Nef had no significant effects on the susceptibility of HIV-1 NL4-3 and CH077 to overexpression of EHMT2, RHOA and CC2D1B in HEK293T cells (Supplementary Fig. 9). Altogether, our results show that "handicapped" TVs lacking specific genes identify innate defense mechanisms that are counteracted by HIV-1 accessory proteins and revealed that inhibition by IFI16 is antagonized by Nef.

## Discussion
In the present study, we exploited the replication fitness of infectious HIV-1 constructs expressing sgRNAs to decipher antiviral mechanisms. We named this technology "Traitor-virus" approach since populations of HIV-1 engineered to express sgRNAs not only allow the pathogen to inactivate antiviral genes (i.e. confer a selective advantage) but also reveal their identity (i.e. the targeted sequence). Each sgRNA represents a unique molecular barcode allowing the association of a selection advantage with a specific cellular gene. Unlike previous methods, this virus-driven technology is highly effective, robust and sensitive because the effect of selective advantages associated with specific sgRNAs is amplified at each round of viral replication. Notably, this closely reflects the impact of fitness advantages during HIV-1 replication in vivo. Competition-based TV screens enable simultaneous evaluation of numerous cellular targets using complex HIV-1-U6-sgRNA-scaffold libraries. Since the readout relies on changes in viral replication fitness and hence changes in the relative frequencies of sgRNAs our approach is highly robust and barely affected by variations in the number of input sgRNA copies. Functional analyses confirmed that TVs identify physiologically relevant cellular factors that restrict HIV-1 replication in primary CD4$^+$ T cells as well as Nef targets.

We generated TVs expressing 1537 different sgRNAs to assess 511 cellular target genes in two different viral backbones and in two Cas9 expressing cell lines. At the end of cell culture passage, sgRNAs targeting *GRN*, *CIITA*, *CC2D1B*, *CEACAM3*, *EHTM2* and *HMOX1* were enriched by ~10- to 500-fold under all selection conditions demonstrating significant selection advantages for the virus. TV-based screens are highly flexible, enabling the monitoring of differences in selective pressures in various cellular environments. Thus, they will allow to elucidate e.g. defense factors in T cells versus macrophages, as well as innate immune mechanisms induced by different types of cytokines. Since inducibility by IFNs is a feature of many restriction factors, we performed the TV screen in the presence and absence of IFN-β. Unexpectedly, most antiviral factors identified were expressed at similar levels and exerted comparable selection pressures under both conditions (Examples shown in Fig. 2). Notably, this was not due to lack of responsiveness of the Cas9 expressing CEM-M7 and SupT1 cells

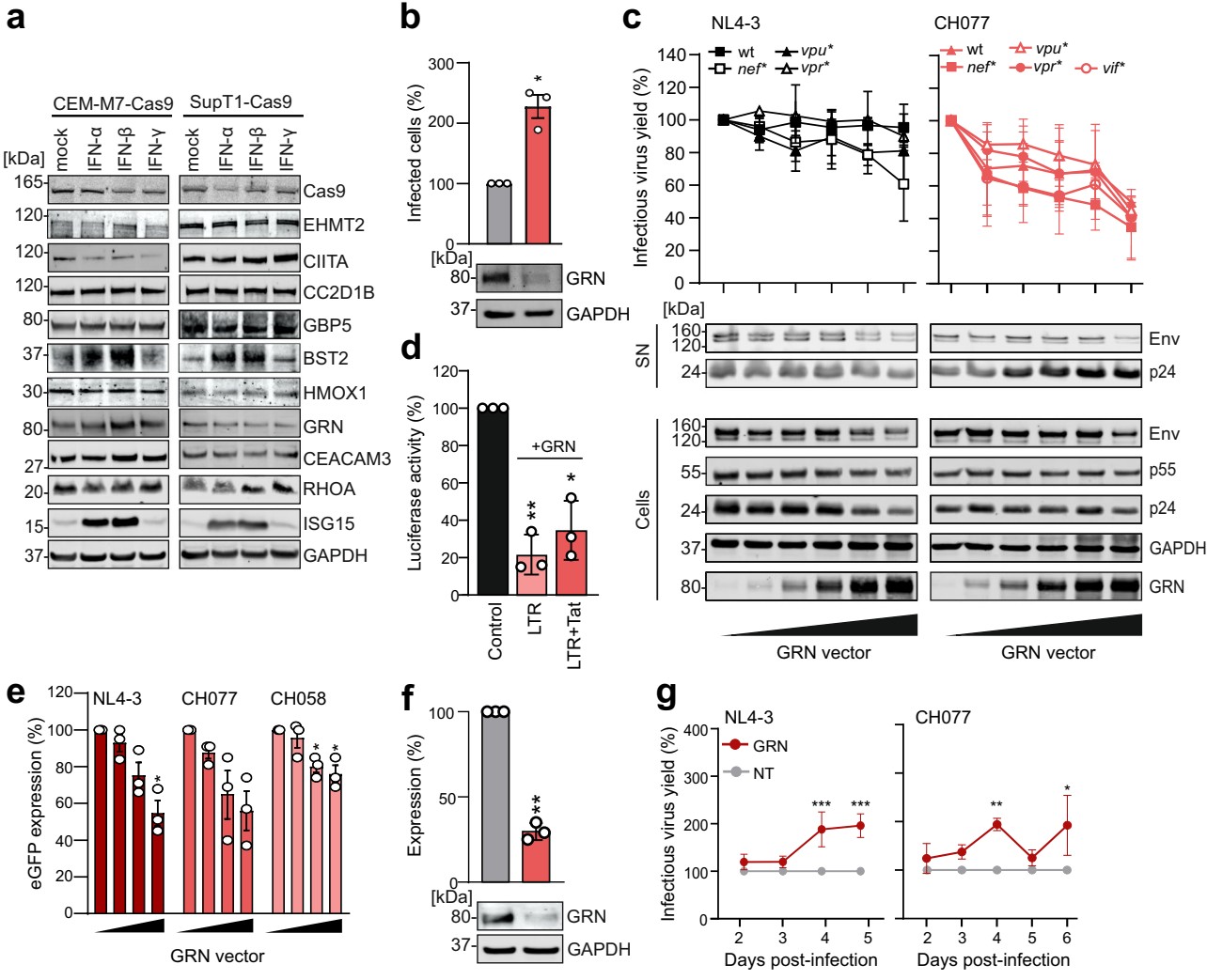

**Fig. 3 | Impact of GRN on HIV-1 replication. a** Immunoblot showing expression of selected cellular factors in CEM-M7 and SupT1-Cas9 cells with or without the indicated IFNs (1000 U/ml). Whole-cell lysates were immunoblotted and stained with antibodies against the indicated proteins. Representative image of $n = 2$ (independent experiments). **b** Percentage of eGFP positive cells indicating infected CEM-M7-Cas9 cells at 4 days post infection electroporated with either the NT (gray) or GRN (red) sgRNA and infected with WT NL4-3. Bars represent the mean of infected cells at 2dpi relative to the control (100%) of $n = 3 \pm$ SEM (independent experiments). (lower panel) Representative WB showing GRN KO efficiency. **c** HEK293T cells were cotransfected with increasing amounts of GRN expression construct and proviral mutants of NL4-3 (black) or CH077 (red) lacking indicated accessory genes. Infectious virus yield was measured using the TZM-bl reporter cell infectivity. Each point represents the mean of $n = 3 \pm$ SEM (independent experiments). (lower panel) Representative WB indicating expression of Env, p55, p24 and GRN in virus supernatants or cell lysates. **d** HEK293T cells were transiently transfected with a luciferase reporter controlled by the HIV-1 LTR and expression constructs for GRN (red) in presence and absence of NL4-3 Tat or a vector control (black). Bars represent the mean of $n = 3 \pm$ SEM (independent experiments). **e** HEK293T cells were transiently transfected with different amount of GRN expression constructs with proviral constructs of NL4-3_eGFP (dark red), CH077_eGFP (red) or CH058_eGFP (light red) as indicated. At 48 h post transfection cells were analyzed by flow cytometry and mean fluorescence intensities (MFI) of eGFP in the eGFP+/GRN+ population relative to vector control (100%) quantified. Bars represent the mean of $n = 3 \pm$ SEM (independent experiments). **f** Representative WB and quantification of GRN KO (red) in primary CD4+ T cells. NT, gray. Bars represent the mean of $n = 3 \pm$ SD (independent donors). **g** CD4+ T cells were electroporated with sgRNA targeting *GRN* (red) or NT (gray), infected with the indicated WT HIV-1 strains and infectious virus yields determined by TZM-bl assays at 2–6 dpi. Dots represent the mean of $n = 6$ (NL4-3) or $n = 3$ (CH077) $\pm$ SEM (independent donors). **b**–**f** Student´s $t$ test with Welch´s correction, two-sided. **g** Two-way Anova with Sidak´s multiple comparison. $*p < 0.05$, $**p < 0.001$, $***p < 0.0001$. Raw $p$ values are provided in Supplementary Data 3.

since expression of ISG15 and Tetherin (BST2) were efficiently induced by IFN-β treatment (Fig. 3a). Thus, our screen identifies antiviral factors that are induced by IFNs, as well as those that are constitutively expressed to confer immediate protection. The latter may represent the real first line of defense as they do not require viral replication and innate immune activation to exert protective effects.

The high sensitivity and experimental setting of the virus-driven approach allows to identify factors that might be missed by current overexpression and KO studies. Overexpression confirmed inhibitory effects of GRN and CC2D1B, while KO of GRN, CIITA and CEACAM3 increased HIV-1 replication in primary CD4+ T cells. It is well known

that overexpression in HEK293T cells is prone to artifacts and manipulation of viral target cells prior to infection, such as in KO settings, may yield misleading results. For example, the CD4 receptor is essential for HIV-1 entry but impairs viral release and infectivity later during the replication cycle[54]. Indeed, our results indicate that CC2D1B may promote viral entry but restrict replication/exit. In addition, KO of some cellular factors (such as RHOA) affects cell proliferation and division precluding meaningful analysis. In the TV approach, HIV-1 itself drives selection and the fitness advantage is determined by the inhibitory effect of the targeted cellular gene. Thus, changes in the abundance of specific sgRNAs in the replicating viral population are a

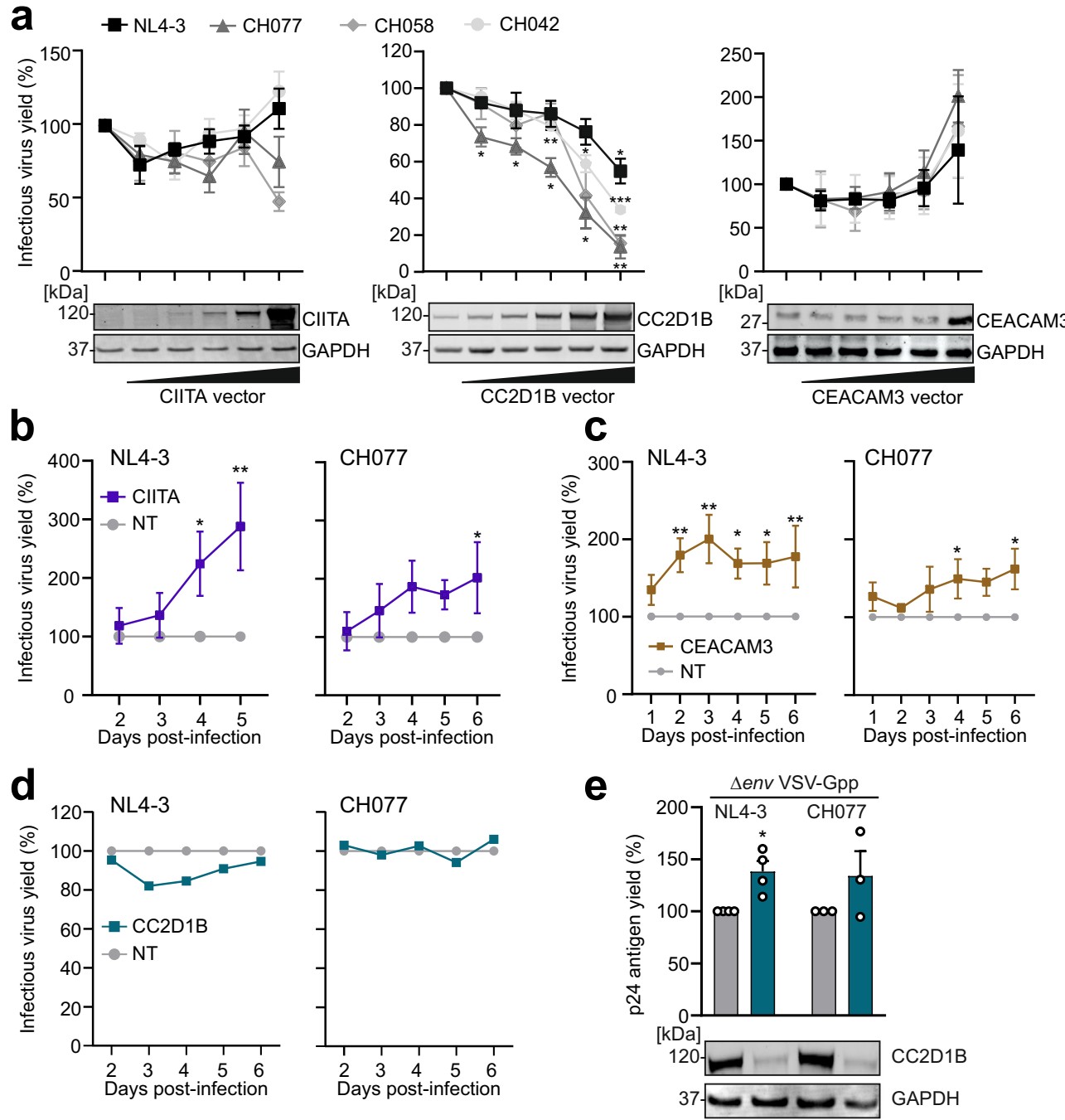

**Fig. 4 | Impact of CIITA, CC2D1B and CEACAM3 on HIV-1 replication.**
**a** HEK293T cells transiently transfected with increasing amount of either CIITA, CC2D1B or CEACAM3 expression constructs and indicated proviral constructs (NL4-3, CH077, CH058, CH042 in different shades of gray). Infectious virus yield was measured using the TZM-bl reporter cell infectivity. Dots represent the mean of three independent experiments ±SEM. **b–d** CD4+ T cells were electroporated with either the sgRNA targeting *CEACAM3* (brown), *CIITA* (purple), *CC2D1B* (petrol), or the NT control, infected with the indicated WT HIV-1 strains and infectious virus yields measured from 2 to 6 dpi by TZM-bl infection assays. Dots represent the

mean of $n = 6$ (**b**) ±SEM, $n = 3$ (**c**) ±SEM or $n = 2$ (**d**) (independent donors). Examples from primary data in Supplementary Figs. 3 and 4. **e** Percentage of p24 antigen in the supernatants of CD4+ T cells electroporated with either the sgRNA targeting *CC2D1B* (petrol) or NT (gray) control at 3 days post infection with VSV-G pseudo-typed Δ*env* NL4-3 or CH077. Bars represent the mean of $n = 4$ (NL4-3) or $n = 3$ (CH077) ± SEM (independent experiments). (lower panel) Representative WB showing CC2D1B KO efficiency. **a**, **e** Student´s *t* test with Welch´s correction, two-sided. **b–d** Two-way Anova with Sidak´s multiple comparison. *$p < 0.05$, **$p < 0.001$, ***$p < 0.0001$. Raw $p$ values are provided in Supplementary Data 3.

robust indicator of the importance of the corresponding antiviral factors.

Many previous screens focused on early steps of the HIV-1 replication cycle[22] and/or analyses of ISGs[23] mainly due to experimental constrains. In comparison, the TV-mediated inactivation of cellular genes is initiated after proviral integration simultaneously with viral

gene expression. Thus, it detects cellular factors presumably affecting viral transcription and latency (IFI16, GRN, CIITA, EHMT2), assembly and release (tetherin, CC2D1B), as well as on virion infectivity (SER-INC5, GBP5) (Supplementary Fig. 10). Notably, EHMT2 is a methyl-transferase that generates H3K9me2, which plays an important role in HIV-1 latency in primary CD4+ T cells[55]. Our results further support that

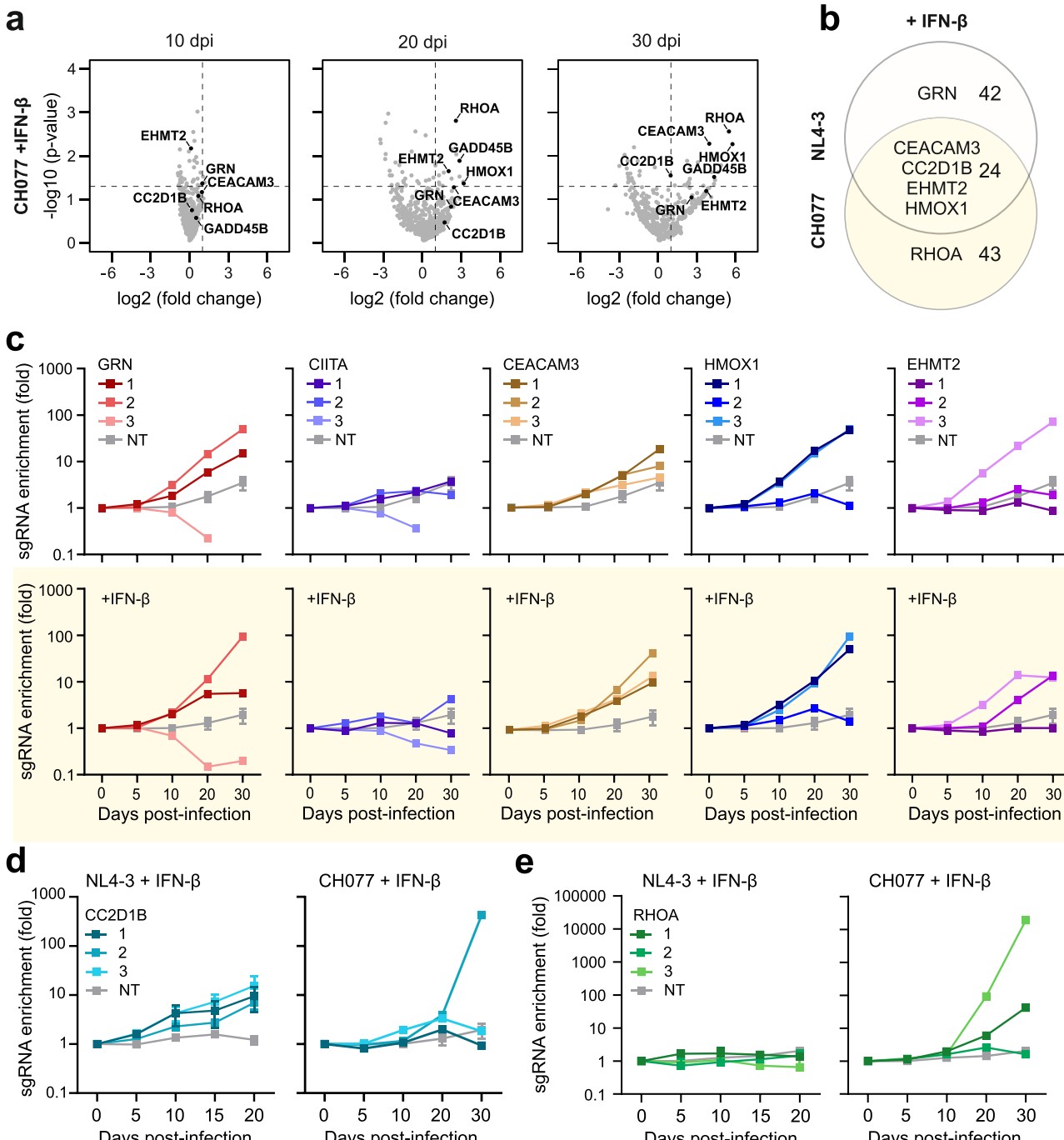

**Fig. 5 | sgRNAs targeting *HMOX1*, *EHMT2*, *CC2D1B* and *RHOA* increase replication fitness of HIV-1 CH077. a** Volcano plots indicating specific target genes of which the sgRNAs are enriched during passage in CEM-M7-Cas9 cells at different days post infection (dpi) after passaging of the TV- CH077-CRF-sgRNA in presence of IFN-β. Dashed lines indicate *p*-value 0.05 and 2-fold change on the Y and X axis, respectively. *P* value, negative binomial (NB) model with robust ranking aggregation (RRA). **b** Venn diagram illustrating the enriched sgRNAs with both viruses in presence of IFN-β. Genes were considered enriched when the MAGeCK score was equal to or above 1.5. **c** Read counts relative to input virus from the MAGeCK analysis showing the enrichment of sgRNAs targeting *GRN* (red), *CIITA* (purple), *CEACAM3* (brown), *HMOX1* (blue) or *EHMT2* (violet) in absence (upper panel) or presence (lower panel) of IFN-β in CEM-M7-Cas9 cells after passaging of the CH077 library. NT, gray. Dots represent *n* = 1. Error bars of the NT sample represents the mean of 11 NT sgRNAs ±SEM. **d** Read counts relative to input virus from the MAGeCK analysis showing the enrichment of sgRNAs targeting *CC2D1B* (turquoise) after passaging the NL4-3 (left) or CH077 (right) library on CEM-M7-Cas9 cells in the presence of IFN-β. NT, gray. Dots represent *n* = 1. Error bars of the NT sample represents the mean of 11 NT sgRNAs ±SEM. **e** Read counts relative to input virus from the MAGeCK analysis showing the enrichment of sgRNAs targeting *RHOA* (green) after passaging the NL4-3 (left) or CH077 (right) library on CEM-M7-Cas9 cells in the presence of IFN-β. NT, gray. Dots represent *n* = 1. Error bars of the NT sample represents the mean of 11 NT sgRNAs ±SEM.

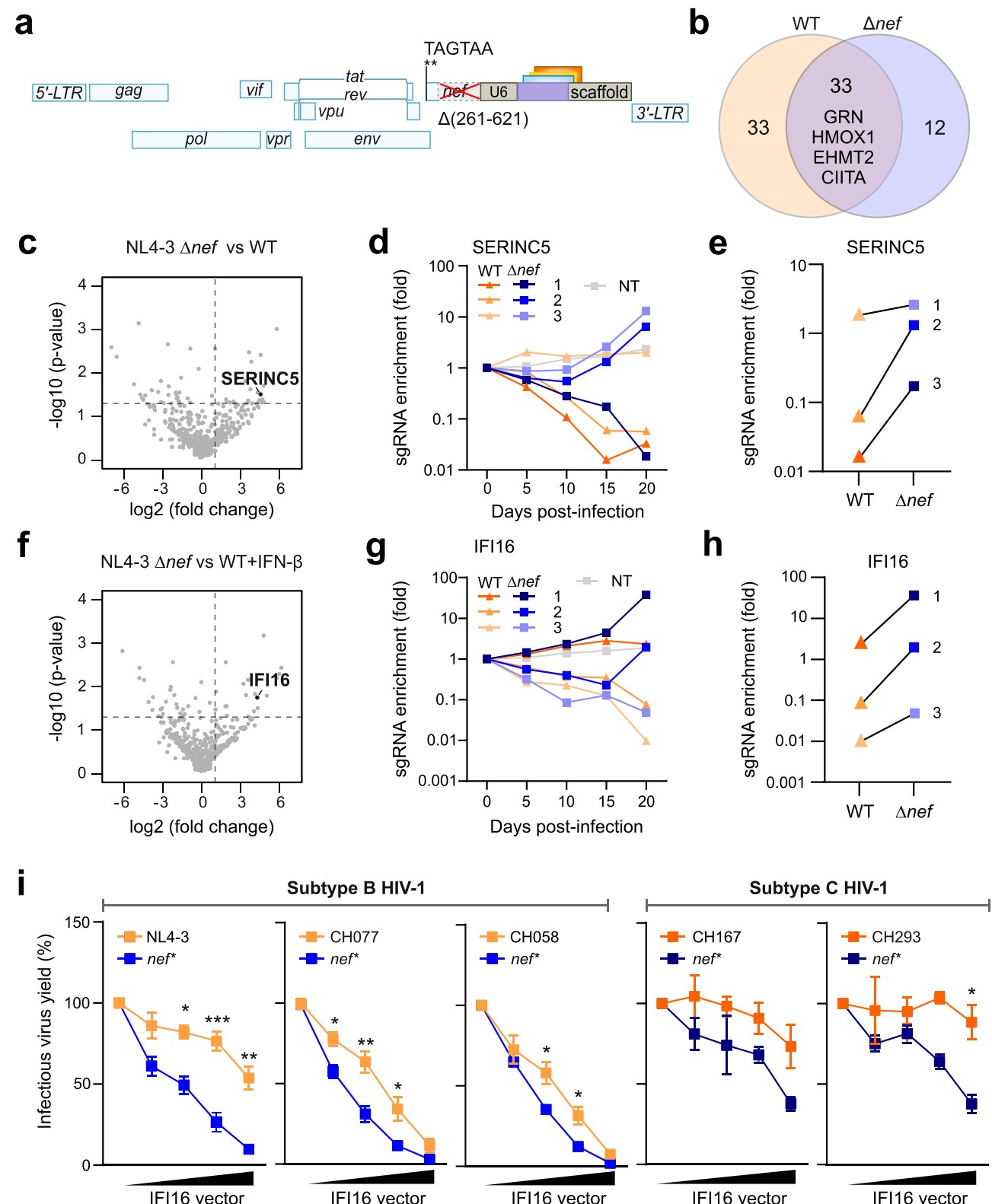

silencing of EHTM2 promotes productive infection and efficient viral transcription. Thus, TV-based screens allow to identify targets for reactivation of latent viral reservoirs representing the major obstacle against a cure of HIV/AIDS[56]. HMOX1 is upregulated in response to oxidative stress and an important anti-inflammatory enzyme. It has been suggested to exert protective effects in HIV-1 infected individuals[57,58] and to restrict SARS-CoV-2[59,60]. We also observed enrichment of sgRNAs targeting *IRF3*, a transcription factor playing a

key role in the induction of innate antiviral defense mechanisms indicating detection of factors setting the cell in an antiviral state rather than inhibiting HIV-1 directly.

In some aspects, the present TV method resembles a recently reported influenza-driven screen for virus dependency factors, which confirmed the attenuating role of TREX1 in viral sensing[61]. While having some similar perks as our system, such as allowing multiple rounds of replication that permit effective detection of fitness advantages, it

**Fig. 6 | Selection of sgRNAs by HIV-1 U6-CRF-sgRNA constructs lacking *nef*.**
**a** Schematic representation of the *nef*-defective HIV-1 TV-NL4-3-CRF-sgRNA constructs. **b** Venn diagram illustrating sgRNAs enriched in presence and absence of Nef. Genes were considered enriched when the MAGeCK score was equal to or above 1.5. **c** Volcano plot indicating sgRNAs targeting *SERINC5* enriched in Δ*nef* kinetics compared to WT kinetic during passage in CEM-M7-Cas9 cells at 15 dpi. Dashed lines indicate *p* value 0.05 and 2-fold change on the Y and X axis, respectively. *P* value, negative binomial (NB) model with robust ranking aggregation (RRA). **d** Read counts relative to input virus from the MAGeCK analysis showing enrichment of individual sgRNAs targeting *SERINC5* in Δ*nef* (blue) and WT (orange)

kinetics. **e** Read counts relative to input virus from the MAGeCK analysis showing enrichment of individual sgRNAs targeting *SERINC5* comparing Δ*nef* (blue) and WT (orange) kinetic at 15 dpi. **f**–**h** sgRNAs targeting *IFI16* as described for *SERINC5* in panels **c**–**e**, except that in enrichment in CEM-M7-Cas9 was determined in the presence of IFN-β and at 20 dpi. **i** HEK293T transiently transfected with increasing amounts of IFI16 expression constructs and indicated proviral WT (orange) or Nef-deficient (*nef**, blue) constructs. Infectious virus yield was measured using the TZM-bl reporter cell infectivity. Dots represents the mean of $n = 3 \pm SD$ (independent experiments). Student´s *t* test with Welch´s correction, two-sided, *$p < 0.05$, **$p < 0.001$, ***$p < 0.0001$. Raw p values are provided in Supplementary Data 3.

relied on artificial induction of factors to identify proviral genes. A loss-of-function approach as in our case has the advantage of identifying cellular factors that affect viral replication at endogenous expression levels. Furthermore, compared to influenza virus, passaging of HIV-1 induces less cytopathic effects and thus loss of cells with increased replication. Most importantly, working with recombinant HIV-1 is established in a plethora of labs worldwide and screening systems based on lentiviruses are highly relevant as commonly usable, flexible and rapidly adoptable tools. In contrast, generation of genetically modified viruses containing segmented negative sense RNA genomes, such as influenza virus, requires technically challenging complex reverse genetics systems[62].

The ease of genetic manipulation of HIV-1-based constructs also allows generation of TVs with specific "handicaps", such as defects in accessory genes, switches in coreceptor tropism or alterations in regulatory elements. For proof of concept, we generated and screened *nef*-deleted NL4-3-based TVs. Predictably, lack of Nef increased the fitness advantage mediated by sgRNAs against *SERINC5*, an established Nef target[50,51] (Fig. 6c–e). Surprisingly, lack of Nef function also increased selection pressure for sgRNAs targeting *IFI16* (Fig. 6f–h). Overexpression analyses confirmed that *nef*-defective primary HIV-1 IMCs are significantly more susceptible to inhibition by IFI16 compared to otherwise isogenic WT viruses (Fig. 6i). IFI16 has been reported to inhibit viral pathogens including HIV-1 by a variety of mechanisms and has also been proposed to play roles in innate sensing of viral pathogens[63–65]. Our finding that the inhibitory activity of IFI16 is not only evaded by an additional NF-κB binding site in the LTR of currently dominating clade C viruses[52] but also counteracted by Nef further supports an important role of this antiviral factor. Our results obtained using otherwise isogenic TV constructs differing in *nef* are proof-of-concept that genetically closely related pairs HIV-1 strains differing in IFN sensitivity and/or accessory gene function will allow to pinpoint factors involved in virus transmission and/or counteracted by Vif, Vpr, Vpu or Nef. In addition, CRISPR/Cas9-based approaches become increasingly versatile. For example, mutated Cas9 allows to enhance gene expression for identification of HIV-1 dependency factors, or Cas12a2 allows targeting of mRNAs instead of cellular genes[66].

Considering the nature of innate immune defenses, most, if not all factors identified in the TV approach will be relevant for other viruses and diseases that involve innate immune responses, as well[2,67]. In fact, characterization of RFs against HIV-1 previously often served as a blueprint to identify important components of cellular defenses, such as APOBEC3, tetherin and SERINC proteins that are now well-known as broad antiviral factors. Our TV approach identified HMOX1, which was previously shown to antagonize SARS-CoV-2[60]. CIITA has been reported to provide cell resistance against Ebola virus and SARS-like coronaviruses[44]. Antiretroviral factors including APOBEC3 and TRIM proteins as well as SAMHD1 also play roles in genomic integrity and cancers[68,69]. Granulin (GRN) is known to be a potent mitogen implicated in many human cancers[70]. These examples indicate that factors identified in TV-approaches are of broad relevance.

In conclusion, we conceived an innovative pathogen-driven screening approach that provides an effective and convenient means

to elucidate which cellular genes affect replication fitness of HIV-1. It is highly versatile and robust and thus will allow to assess zoonotic potential, degree of adaptation to human and/or the repertoire of their accessory genes to obtain exciting insights into the complex virus-host pathogens and defense mechanisms against viral pandemics. We present a focused screen but high cloning efficiencies and infectious virus titers offer the possibility for genome-wide unbiased identification of antiviral factors and comprehensive elucidation of complex virus-host interactions.

## Methods
### Cell culture
All cells were cultured at 37 °C in a 5% $CO_2$ atmosphere. Human embryonic kidney 293T cells (HEK293T; ATCC) and TZM-bl cells were maintained in Dulbecco's Modified Eagle Medium (DMEM) supplemented with 10% heat-inactivated fetal calf serum (FCS), L-glutamine (2 mM), streptomycin (100 µg/ml) and penicillin (100 U/ml). TZM-bl cells were provided and authenticated by the NIH AIDS Reagent Program, Division of AIDS, NIAID, NIH from Dr. John C. Kappes, Dr. Xiaoyun Wu and Tranzyme Inc. TZM-bl are derived from HeLa cells, which were isolated from a 30-year-old female. CEM-M7-Cas9 and SupT1 CCR5 high Cas9 (SupT1-Cas9) cells were cultured in Roswell Park Memorial Institute (RPMI) 1640 Medium supplemented with 10% heat-inactivated fetal calf serum (FCS), L-glutamine (2 mM), streptomycin (100 µg/ml) and penicillin (100 U/ml).

### Primary cell cultures
PBMCs from healthy human donors were isolated using lymphocyte separation medium (Biocoll separating solution; Biochrom) or lymphoprep (Stemcell). CD4+ T cells were negatively isolated using the RosetteSep Human CD4+ T Cell Enrichment Cocktail (Stem Cell Technologies # 15061) or the EasySep Human Naïve CD4+ T cell Isolation Kit (Stem Cell Technologies #17953) according to the manufacturer's instructions. Primary cells were cultured in RPMI-1640 medium containing 10% FCS, glutamine (2 mM), streptomycin (100 µg/ml), penicillin (100 U/ml) and interleukin 2 (IL-2, Miltenyi Biotec #130-097-745) (10 ng/ml).

### Ethics
The use of human PBMCs was approved by the Ethics Committee of the Ulm University Medical Center (Approval 93/21-FSt/TR). All donors were anonymized prior to the experiments and are randomly chosen from a pool of healthy donors below 30 years. Informed written consent was given and no compensation provided. Sex and/or gender were not considered for the study design and were determined based on self-report.

### Expression constructs
Expression vectors for GRN, CIITA, CC2D1B, RHOA, CEACAM3 were purchased from GenScript (#OHu25975C, #OHu21123C, #OHu11655C, # OHu26883C, # OHu15558C). The expression vector for IFI16 was previous described[53]. Constructs expressing the HIV-1 NL4-3 LTR and HIV-1 proviral constructs co-expressing eGFP via an IRES were generated previously[53].

## Generation of cell lines constitutively expressing Cas9

To generate lentiviral backbones constitutively expressing Cas9, the ORF of Cas9 (humanized *S. pyogenes* Cas9 sequence) was fused to a nuclear localization signal and cloned behind a CMV promoter flanked by 5′LTR and 3′LTR sequences derived from HIV-1. Third generation lentiviral particles were produced by complementing the backbone in HEK293T cells with VSV-G, HIV-1 Gag/Pol and HIV-1 Rev expression vectors. CD4+ T-cell lines (CEM-M7 and SupT1) were transduced with the lentiviruses using spinoculation. 72 h post transduction the cells were selected using 10 μg/ml of Blasticidin. After the cells recovered for one-week, single cells were sorted in 96 well plates. Three weeks post sorting, single clone colonies were harvested and screened for Cas9 expression via Western blot analysis.

## Construction of sgRNA library based on full length HIV-1

To generate the HIV-1 backbones, the sgRNA cassette carrying the human U6 promoter and the invariant scaffold sgRNA sequence were inserted into the HIV-1 NL4-3 and HIV-1 CH077 proviral DNA between separated *nef* and 3′LTR region using homologous recombination (NEB builder Hifi DNA assembly mastermix, NEB #E2621). The U6 promotor and the invariant scaffold are separated by a unique BsmBI restriction site using Q5 Site-Directed Mutagenesis Kit (NEB #E0554). Additional BsmBI restriction sites in the HIV-1 sequence were removed using Splicing by overlap extension (SOE) PCR (Forward Primer (Q5_del_BsmBI_CH077_F): 5′-AGCTCCCGGAcACGGTCACAG-3′, Reverse Primer (Q5_del_BsmBI_CH077_R): 5′-GCATGTGTCAGAGGTTTTCAC-3′, Forward Primer (Q5_del_BsmBI_NL4.3vec_F): 5′-CTGTGACCGTgTCCGGGAGCT-3′, Reverse Primer (Q5_del_BsmBI_NL4.3vec_R): 5′-CTTGTCTGTAAGCGGATGCC-3′, Forward Primer (Q5_del_BsmBI_NL4.3nef_F): 5′-AAAGAATGAGgCGAGCTGAGC-3′, Reverse Primer (Q5_del_BsmBI_NL4.3nef_R): 5′-AAAGAATGAGgCGAGCTGAGC-3′. The nucleotide sequence of *nef* was codon optimized to avoid recombination with the 3′ LTR: The fragment was synthetized by Twist Bioscience and cloned into the corresponding proviral DNA using XhoI/MluI (NL4-3) and KflI/MluI (CH077). For the generation of a small targeted library (BST2, GBP5, NT) oligonucleotides were purchased from Biomers and designed with flanking regions in 3′ (3′-GTGGAAAGGACGAAACACCG-5′) and 5′ (3′-GTTTTAGAGCTAGAAA-TAG-3′) of the sgRNA overlapping with the backbone sequence to facilitate insertion by homologous recombination ((gRNA-GBP5-1): 5′-ACAATCGCTACCACAACTAC-3′, (gRNA-GBP5-2): 5′-ATTAGTTCTGCTTGACACCG-3′, (gRNA-BST2-1): 5′- CTGGATGCAGAGAAGGC-CCA-3′, (gRNA-BST2-2): 5′- CTCTTCTTAGATGGCCCTAA-3′, (gRNA-NT): 5′-ACGG-AGGCTAAGCGTCGCAA-3′). For the generation of the library targeting 511 genes, a pool of amplicons containing individual sgRNAs was purchased from Twist Bioscience. The variable sgRNA targeting sequences (18 nucleotides) were taken from the Gecko v2 library (3 for each gene). We selected sgRNAs targeting 511 cellular genes sharing features of know restriction factors or proposed to play roles in HIV-1 infection[24,36]. To insert the sgRNA targeting sequences, the proviral backbones were linearized by using BsmBI. Recombination was performed by incubating the linearized vector (330 ng) with the amplicons pool (50 ng) and the NEBuilder HiFi DNA Assembly (NEB #E2621) 50 °C for 15 min to one hour. Afterwards, the reaction was purified using the Monarch DNA Gel Extraction Kit (NEB #T1020L) and transformed by electroporation using the Gene Pulser Xcell (1700 V, 25 μF, 200 Ω, 1 pulse, Biorad) in C2989 5alpha electrocompetent bacteria (NEB, #C3020K). After the bacteria recovered for one hour at 37 °C in SOC medium, they were plated on 6 15 cm agarose dishes and incubated at 30 °C for 40 h. All colonies were collected by scraping and the DNA was extracted using the Plasmid maxiprep (Qiagen #12165). For a small proof-of-principle library containing 7 different targets, sgRNA integration and complexity of the library was quantified using SYBR-green qPCR (SYBR™ Green PCR

Master Mix, Applied Biosystems #4309155) with one forward primer binding the U6 promoter region and specific reverse primers for each sgRNA. To generate NL4-3 TVs mutants lacking the *nef* gene, we introduced a stop codon at the beginning *nef* and subsequentially we deleted 360 nucleotides (from nucleotide 261 to 621, Stop Codon NL4-3 Nef Forward: 5′-CTATAAGATG-TAGTAAAAGTGGTCAAAAAGTAGTG-3′, Stop Codon NL4-3 Nef Reverse: 5′-CAAAA-TCCTTTCCAAGCC-3′, Nef deletion Forward: 5′-ACGCGTCCAAGGTCGGGC-3′, Nef deletion Reverse: 5′-AGATCTACAGCTGCCTTGTAAGTCATTGG-3′) using Q5 Site-Directed Mutagenesis Kit (NEB #E0554S).

## Verification of viral recombination by PCR

To check for recombination and loss of the cassette during passaging, viral RNA was isolated at different time points with the QIAamp Viral RNA Mini Kit (Qiagen # 52906). cDNA was synthetized using the PrimeScript RT Reagent Kit (Takara #RR037A) according to the manufacturer's instructions. The cassette was amplified using flanking primers (Forward Primer (PCR_Recombination_F): 5′-GTGGA-ACTTCTGGGA-3′, Reverse Primer (PCR_Recombination_R): 5′-ACTGCTAGAGATTTT-CCACACTGACTAAAAG-3′. PCR reactions were loaded onto a 1% agarose and ran at 140 V for 30 min.

## Stimulation with type I and II interferons

One million CEM-M7-Cas9 or SupT1-Cas9 cells were seeded in 1 mL RPMIXXX in 12-well plates. Cells were stimulated with IFN-α (500 U/ml, R&D systems 11100-1), IFN-β (500 U/ml, R&D systems 8499-IF-010) or IFN-γ (200 U/ml, R&D systems 285-IF-100). 24 h post-stimulation whole cell lysates were generated.

## CRISPR/Cas9 KO in T cells

CD4+ T lymphocytes were isolated from healthy donors as described above. Cells were stimulated with IL-2 (10 ng/ml) (Miltenyi Biotec #130-097-745) and with anti-CD3/CD28 beads (Gibco #11132D) for 3 days. Cells were cultured in RPMI-1640 medium containing 10% FCS and IL-2 (10 ng/ml). $1 \times 10^6$ primary CD4 + T cells (stimulated) or $1 \times 10^6$ CEM-M7-Cas9 cells were transfected with the HiFi Cas9 Nuclease V3 (IDT)/gRNA complex (80 pmol/300 pmol) (Lonza) using a non-targeting or a GRN(5′-GCGATCCTGCTTCCAAAGATC-3′), CIITA (5′-GCCCCTAGAAGGTGGCTACC-3′), RHOA (5′-TATCGAGGTGGATGGAAAGC-3′), CC2D1B (5′-GAGTTGGCGGCAGACTGTATG-3′), CEACAM3(5′-GTGTCTCTCGACCGCTGTTTG-3′)-specific sgRNAs or NT control (5′-ACGGAGGCTAAGCGTCGCAA-3′), using the Amaxa 4D-Nucleofector Human Activated T Cell P3 Lonza Kit (Lonza #V4XP-3032), pulse code E0115. At four- and three- days post Cas9/sgRNA-transfection respectively, 1 million cells/sample were infected with the indicated HIV-1 strains by spinoculation. From 2 to 5 or 6 dpi, supernatants were harvested and infectious virus yield via the TZM-bl reporter cells assay.

## Transfection and production of viral stocks

HEK293T cells were transiently transfected using TransIT-LT1 (Mirus #MIR2306) according to the manufacturer's protocol at a ratio of 3 μL of transfection reagent per 1 μg of DNA and the medium was replaced 24 h post transfection. To test the antiviral effect of potential restriction factors, pcDNA-based expression constructs cotransfected with the proviral constructs. Whenever different amounts of pcDNA expression vectors were used within an experiment, empty vector control plasmids were used to keep the total DNA amount constant for all samples. The transfected cells were incubated for 8–16 h before the medium was replaced by fresh supplemented DMEM. To generate virus stocks, one day before transfection, 10 mio cells were seeded in 15 cm dishes in 20 ml medium to obtain a confluence of 70–80% at the time of transfection. For transfection, 25 μg of DNA was mixed with 75 μl LT1, incubated 20 min at RT and added dropwise to the cells. 48 h post transfection, the virus was harvested, centrifuged 5 min at 2000rpm and concentrated 10 times using Amicon Ultra 15 mL Filters

(Merck #UFC910096). The concentrated virus aliquoted and stored at −80 °C.

## VSV-G-pseudo-typed HIV-1

To generate VSV-G-pseudo-typed HIV-1, HEK293T cells were transiently transfected using the calcium-phosphate precipitation method[53]. Briefly 5 µg of proviral DNA and 1 µg of expression plasmids for VSV-G was mixed with 13 µl 2 M CaCl$_2$ and filled up with water to 100 µl. Afterwards, 100 µl of 2× HBS was added dropwise to this mixture, which was mixed by pipetting and added dropwise to the cells seeded in 6 well plates.

## Infection, kinetics and traitor virus enrichment

To start the replication kinetic, 1 million cells were infected with the indicated HIV-1 library constructs via spinoculation (2 h at 26 °C). Afterwards cells were washed three times with RPMIxxx and seeded in 6 well plates at a cell density of 1 million/ml. Every two to three days, infection was monitored by flow cytometry (see below). When infection was higher than 20%, it was reduced to 1% for the next 2 days by addition of uninfected cells. From 5dpi, cells were treated with IFN-β (R&D Systems #8499-IF-010, 1000U/ml for CEM-M7-Cas9 and 100 U/ml for SupT1-Cas9) and IFN-β was refreshed every three days.

## Viral RNA preparation for sequencing

Viral RNA levels were determined in supernatants collected from HIV-1 infected cells at 5, 10, 15, 20, 30- and 40-days post-infection. Total RNA was isolated using the Viral RNA Mini Kit (Qiagen) according to the manufacturer's instructions. cDNA reactions were performed according to the manufacturer's instructions of the PrimeScript RT Reagent Kit (Takara) using primers specifically targeting the U6 and scaffold region (forward primer 5´-CCGACTCGGTGCCACTTTTT-3´, reverse primer 5´-CGTGACGTAGAAAGTAATAATTT-CTTGGG-3´). cDNA reactions were purified using the Monarch PCR Purification Kit (NEB #T1030L) and eluted in 10 µl elution buffer. The sgRNA cassette was amplified using the NEBNext® High-Fidelity 2X PCR Master Mix (NEB) and primers including Illumina adapters and 8nt barcodes to allow Next Generation Sequencing analysis (Supplementary Data 2). PCR reactions were purified using the Monarch PCR Purification Kit (NEB # T1030L) and eluted in 10 µl elution buffer.

## Next generation sequencing

NGS was performed using the Illumina NextSeq2000 platform with 60 base-pair paired-end runs. Raw reads were demultiplexed on the Galaxy version 23.0 platform, forward and reversed reads were merged with SeqPrep 0.2.2 and aligned to the custom library sequences using the MAGeCK algorithm suite (Version 0.5.9.2.4). Individual read counts are determined and median-normalized to for the effect of library sizes and read count distributions. Individual sgRNAs targeting the same gene are summarized, and a variance model calculated using a negative binomial model to statistically assess the difference between control (input) and the conditions (different days). Targets are ranked by MAGeCK according to their *p*-value via a modified robust ranking aggregation (RRA) algorithm (α-RRA) to identify enriched genes. Overrepresented sgRNA sequences compared to the input control represent viruses carrying a sgRNA targeting a gene, that restricts viral replication. Volcano plots were generated using R version 4.1.1 and ggplot2 version 3.3.5.

## Venn diagrams

Lists of enriched genes were generated for each condition by selecting genes based on the positive MAGeCK score. Genes were considered enriched when the -log$_{10}$ of the positive score was above 1.5. Genes overlap was calculated using the bioinformatics tool from UGent (https://bioinformatics.psb.ugent.be/webtools/Venn/).

## SYBR green qRT-PCR

To determine the relative enrichment of *GBP5* and *BST2* sgRNAs over time compared to the NT, we performed qRT-PCR using the SYBR Green PCR Master Mix (Applied Biosystems #A25742) following the manufacturer protocol. In brief, we diluted the cDNA and we perform the RT-qPCR reactions using specific primers flanking the sgRNAs regions (U6 Forward_SYBR: 5´-AGAATTAATTTGACTGTAAACACAAAG ATATTAG-3´, GBP5-1grRNA Reverse_SYBR: 5´-CGTAGTTGTGGTAGCG ATTGT-3´, GBP5-2 sgRNA Reverse_SYBR: 5´-CGGTGTCAAGCAGAA CTAAT-3´, BST2-1 sgRNA Reverse_SYBR: 5´-GGCCTTCTCTGCATCC AG-3´, BST2-2 sgRNA Reverse_SYBR: 5´-AACTTAGGGCCATCTAAG AAGAG-3´, NT sgRNA Reverse_SYBR: 5´-TTGCGACGCTTAGCCTC-3´). Values were normalized on the values from 3 days post infection.

## Supernatants and whole cell lysates

To determine expression of cellular and viral proteins, cells were washed in PBS and subsequently lysed in Western blot lysis buffer (150 mM NaCl, 50 mM HEPES, 5 mM EDTA, 0.1% NP40, 500 µM Na$_3$VO$_4$, 500 µM NaF, pH 7.5) or radioimmunoprecipitation assay (RIPA) buffer (50 mM Tris-HCl; pH 7.4, 150 mM NaCl, 1% (v/v) NP-40, 0.5% (w/v) deoxycholic acid (DOC), 0.1% (w/v) SDS) supplemented with protease inhibitor (Roche, 1:500). After 5 min of incubation on ice, samples were centrifuged (4 °C, 20 min, 12,000 × *g*) to remove cell debris. The supernatant was transferred to a fresh tube, the protein concentration was measured with Pierce Rapid Gold BCA Protein Assay Kit (Thermofisher) and adjusted using Western blot lysis buffer. Supernatants were centrifuged on top of a 20% sucrose layer in at 21,000 × *g* for 2 h. The viral pellet was then lysed in Western blot lysis buffer with 4x Protein Sample Loading Buffer (LICOR) supplemented with 10% β-mercaptoethanol (Sigma Aldrich) and heated at 95 °C for 5 min.

## SDS-PAGE and immunoblotting

Whole cell lysates were mixed with 4× Protein Sample Loading Buffer (LI-COR, at a final dilution of 1×) supplemented with 10% β-mercaptoethanol (Sigma Aldrich), heated at 95 °C for 5 min, separated on NuPAGE 4 ± 12% Bis-Tris Gels (Invitrogen) for 90 min at 100 V and blotted onto Immobilon-FL PVDF membranes (Merck Millipore)[71]. The transfer was performed a constant voltage of 30 V for 30 min using semi-dry transfer system. For larger proteins (Cas9, EHMT2), transfer was performed at a constant Amperage 0,4 A for 2 h using a wet transfer system. After the transfer, the membrane was blocked in 1% Casein in PBS (Thermo Scientific). Proteins were stained using primary antibodies against GRN (Abcam #ab208777, 1:200), CIITA (Santa Cruz #sc-13556, 1:200), EHMT2 (Cell Signaling #3306, 1:200), CC2D1B (Proteintech #20774-1-AP, 1:500), HMOX1 (Sigma MA1-112, 1:200), BST2 (Proteintech 13560-1-AP, 1:500), GBP5 (Santa Cruz #sc-1603539, 1:200), CEACAM3 (Abcam #ab196606, 1:200), ISG15 (Santa Cruz #sc-166755, 1:200), IFI16 (Santa Cruz #sc-8023, 1:150), RHOA (Abcam #ab54835, 1:200), GAPDH (Biolegend #607902, 1:1000), Cas9 (Cell Signaling #14697, 1:1000), HIV-1 p24 (Abcam #6604667, 1:1000) HIV-1 Env (NIH AIDS Reagents program #ARP-12559, 1:1000), HIV-1 Nef (NIH AIDS Reagents program #ARP-1539, 1:500) and Infrared Dye labeled secondary antibodies (IRDye 680RD Goat anti-Rabbit IgG (H + L), LI-COR #926-68071, 1:10,000; IRDye 800CW Goat anti-Mouse IgG (H + L), LI-COR #926-32210, 1:10,000; IRDye 800CW Goat anti-Rabbit IgG (H + L), LI-COR #926-32211, 1:10,000; IRDye 800RD Goat anti-Rat IgG (H + L), LI-COR #925-32219, 1:10,000; IRDye 680RD Goat anti-Rat IgG (H + L), LI-COR #926-68071, 1:10,000). Band intensities were quantified using Image Studio (LI-COR). Uncropped and unprocessed scans are provided in the source data file.

## MTT assay

To determine the metabolic activity of the cells, we performed the MTT assay where 0.5 mio CD4[+] T cells were seeded in a 96 well plate. Afterwards, 10 µl of MTT solution was added to the cells culture (1:10

diluted with PBS). 3 h later, 200 µl of DMSO and EtOH mixture (1:1) was added and the plates were measured at the luminometer (absorbance of 493 nm with the baseline correction at 650 nm).

### CellTiter-Glo
To test the cell viability, CellTiter-Glo Luminescent Cell Viability Assay (Promega) was performed according to the manufacturer's protocol. Briefly, 0.5 mio primary CD4$^+$ T cells were lysed with 300 µl of 5× passive lysis buffer, and 25 µl was transferred into a fresh plate in duplicates. Twenty-five microliters of the CellTiter-Glo Reagent was added to the lysates and incubated for 5 min at room temperature. Luminescent signal was read using an Orion microplate luminometer (Berthold).

### Flow cytometry
To monitor infection during the replication kinetic, flow cytometry was used to quantify the infected cells. For CEM-M7-Cas9 kinetics, ~400,000 cells were harvested, washed once with PBS and stained for 15 min at RT in the dark with eBioscience Fixable viability dye 780 (ThermoFisher Scientific, 1:1000 in PBS). Cells were washed twice with PBS and fixed in 2% PFA for 30 min at 4 °C. For SupT1-Cas9 kinetic and to monitor KO efficiencies in cells infected with HIV-1 either carrying the NT or BST2 or GBP5 sgRNA, ~400,000 cells were harvested, washed once with PBS and stained for 30 min at RT in the dark with anti-CD4 antibody (PerCP-Cy5.5, Biolegend #317428, 1:50 in PBS) and eBioscience Fixable viability dye 780 (ThermoFisher Scientific #65-0865-14, 1:1000 in PBS). Afterwards, cells were washed twice with PBS and permeabilized 20 min with 200 µl BD Cytofix/Cytoperm Fixation/Permeabilization Solution Kit (BD Biosciences) at RT. Cells were washed twice with 200 µl 1X Perm/Wash solution and stained 1 h at 4 °C with anti-HIV-1 p24 (RD1/PE, Beckman Coulter #6604667, 1:100 in 1X Perm/Wash solution) or anti-BST2 (Proteintech #13560-1-AP, 1:100 in 1X Perm/Wash solution) or anti-GBP5 (Santa Cruz #sc-1603539, 1:100 in 1X Perm/Wash solution). After washing twice with 1X Perm/Wash solution, wells were either stained with secondary antibody goat anti rabbit (PE, Abcam ab97070, 1:100 in 1X Perm/Wash) or fixed in 2% PFA for 30 min at 4 °C. After 1 h at 4 °C, cells stained with secondary antibodies were washed twice with 200 µL 1X BD Perm/Wash solution and fixed with 2% PFA for 30 min at 4 °C. To monitor activation state, CD4 + T cells were stained for 45 min at RT with eBioscience Fixable viability dye 780 (ThermoFisher Scientific, 1:1000 in PBS), anti-CD25 (FITC, BD Pharmigen #555431, 1:5 in PBS) and anti-HLA-DR (PE-Cy5, BD Pharmigen #555813, 1:5 in PBS). Cells were washed twice with PBS and fixed 2% PFA for 30 min at 4 °C. Cells were washed twice with PBS and fixed 2% PFA for 30 min at 4 °C. To measure CEACAM3 KO, CD4 + T cells were stained for 45 min at RT with eBioscience Fixable viability dye 780 (ThermoFisher Scientific, 1:1000 in PBS) and anti-CD66d/e (Alexa Fluor 647, Biolegend #392806, 1:100 in PBS) or isotype control (Alexa Fluor 647, Santa Cruz #24636, 1:40 in PBS). Cells were washed twice with PBS and fixed 2% PFA for 30 min at 4 °C. Cells were acquired with BD FACSCanto II Flow Cytometer (BD Biosciences).

### Effects of GRN and CIITA on LTR-driven eGFP expression
HEK293T cells were co-transfected with expression constructs for GRN or CIITA and HIV-1 NL4-3, CH077 and CH058 proviral constructs co-expressing eGFP via an IRES. 48 h after transfection cells were harvested, washed in 500 µl PBS and stained for 15 min at RT in the dark with eBioscience Fixable viability dye 780 (ThermoFisher Scientific #65-0865-14, 1:1000 in PBS). Afterwards cells were permeabilized 20 min with 200 µl BD Cytofix/Cytoperm Fixation/Permeabilization Solution Kit (BD Biosciences)at RT, afterwards washed twice with 200 µl BD 1X Perm/Wash solution. Cells were stained with anti-GRN (Abcam #ab208777, 1:100 in 1X Perm/Wash solution) or anti-CIITA (AF647, Santa Cruz #sc-13556, 1:100 in 1X Perm/Wash solution) for 1 h at 4 °C. After washing twice with 200 µl 1X Perm/Wash cells stained with conjugated CIITA antibody were fixed in 2% PFA for 30 min at 4 °C.

Cells stained with GRN antibody were stained 1 h at 4 °C with secondary antibody goat anti rabbit (PE, Abcam #ab97070, 1:100 in 1X Perm/Wash). Cells were washed twice with 200 µL 1X Perm/Wash solution and fixed in 2% PFA for 30 min at 4 °C. Cells were acquired on FACSCanto II Flow Cytometer (BD Biosciences). Mean fluorescence intensities (MFI) of eGFP in the GRN + /eGFP+ or CIITA + /eGFP+ population was determined.

### Viral promoter activity
To determine the effect of GRN on the activity of different viral promoters, 135,000 HEK293T cells/well were seeded in 24 well plates. Cells were cotransfected with firefly luciferase reporter constructs (5 ng) under the control of the HIV-1 LTR or the CMV IE promoter and expression constructs for GRN (50 ng) or a vector control using the calcium phosphate method. In some cases, expression constructs for HIV-1 NL4-3 Tat (500 pg) were cotransfected to activate the LTR promoter. 40 h post-transfection, cells were lysed and firefly luciferase activity was determined.

### Luciferase assay
To determine LTR-driven expression, the cells were lysed in 300 µl of Luciferase Lysis buffer (Promega #E1531) and firefly luciferase activity was determined using the Luciferase Assay Kit (Promega #E1501) according to the manufacturer's instructions on an Orion microplate luminometer (Berthold).

### Viral infectivity
To determine infectious virus yield, 10,000 TZM-bl reporter cells/well were seeded in 96-well plates and infected with cell culture supernatants in triplicates on the following day. Three days post-infection, cells were lysed and *β-galactosidase* reporter gene expression was determined using the X-GalScreen Kit (Applied Bioscience #T1027) according to the manufacturer's instructions with an Orion microplate luminometer (Berthold).

### ELISA p24 and virion infectivity analysis
HIV-1 p24 amounts in cell culture supernatants were determined using an in-house ELISA. Briefly, 96-well MaxiSorp microplates (Sigma) were coated with 0.5 mg/ml anti-HIV-1 p24 (EXBIO #11-CM006-BULK) and incubated in a wet chamber at RT overnight. The plates were then washed 3 times with PBS-T (PBS and 0.05% Tween 20) and incubated with blocking solution (PBS and 10% (v/v) FCS) for 2 h at 37 °C. After washing, the plates were loaded with 100 µL serial dilution of HIV-1 p24 protein (Abcam #ab43037) as standard and dilutions of virus supernatants lysed with 1% (v/v) Triton X-100 and incubated overnight in a wet chamber at RT. After washing unbound capsid, 100 µl/well polyclonal rabbit antiserum against p24 antigen (Eurogentec, 1:1,000 in PBS-T with 10% (v/v) FCS) was added for 1 h at 37 °C. After washing, 100 µL of goat anti-rabbit HRP-coupled antibody (Dianova #111-035-008, 1:2000) was loaded on the plates and incubated for one hour at 37 °C. Finally, the plates were washed and 100 µL SureBlue TMB 1-Component Microwell Peroxidase Substrate (Medac #52-00-04) was added. After 20 min shaking at 450 rpm and RT, the reaction was stopped with 0.5 M H$_2$SO$_4$ (100 µl/well). The optical density was determined by comparing with a standard curve and measured at 450 nm and 650 nm with the Thermo Max microplate reader (Molecular devices).

### Statistics
Statistical analyses were performed using GraphPad PRISM 10 (GraphPad Software). *P* values were determined using a two-tailed Student's *t* test with Welch's correction or Two-way Anova with Sidak´s multiple comparison. Unless otherwise stated, data are shown as the mean of at least three independent experiments ±SEM. Significant differences are indicated as: *$p < 0.05$; **$p < 0.01$; ***$p < 0.001$. Statistical parameters are specified in the figure legends.

## Reporting summary

Further information on research design is available in the Nature Portfolio Reporting Summary linked to this article.

## Data availability

The next generation sequencing data generated in this study have been deposited in the NCBI's Gene Expression Omnibus database under accession code GSE245526 including raw sequencing data files and processed files. All other data are available in the main text or the supplemental information. Source data are provided with this paper.

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

## Acknowledgements

We thank Dré van der Merwe, Johannes Lang, Regina Burger, Jana-Romana Fischer, Birgit Ott, Daniela Krnavek and Martha Mayer for technical assistance and Christina Stürzel for help with the generation of proviral constructs. Caterina Prelli Bozzo, Alexandre Laliberté and Chiara Pastorio are part of the International Graduate School for Molecular Medicine Ulm (IGradU). This study was supported by an ERC Advanced grant (Project 101054456, Traitor-Viruses) to F.K. and German Research Foundation (DFG) grants (CRC 1279, SPP 1923 to F.K. and K.M.J.S.; SP 1600/7-1, SP 1600/9-1 to K.M.J.S.) and a German Federal Ministry of Education and Research (BMBF) junior group to K.M.J.S. (IMMUNOMOD-01KI2014).

## Author contributions

C.P.B. and A.L. performed most experiments with support by A.D.L., C.P., K.R. and M.V. F.K. and K.M.J.S. conceived the study and planned experiments. S.K, A.G. and H.B. performed the deep-sequencing ana-lyses. F.K. wrote the initial draft of the manuscript. All authors reviewed and approved the manuscript.

## Funding

## Competing interests

The authors declare no competing interests.
