## [Peer Review File · Nature Communications]

REVIEWER COMMENTS

Reviewer #1 (Remarks to the Author):

This paper reports a construction of a pool of HIV-1 genomes encoding sgRNAs targeting candidate restriction factors, and demonstrates the outgrowth of selected members of the library upon passage in permissive cells. (The genome needed to be engineered to prevent excision of the sgRNAs by recombination.) The benefit over earlier screens using single-cycle reporters is the ability to detect genes with weaker activity, and acting at late stages of the life cycle. (Note: it may miss genes acting very early, before the sgRNAs have acted.) New genes are identified here.

The most highly enriched sgRNAs targeted six particular host genes chosen for further study. The retest of overexpression or KO of these resulted in varying effects on virus replication, with several having moderate impact on replication in cell lines and primary CD4+ cells – some effects as expected, some not. Repeats were performed with alternative viral strains and host cells, with some differences, and with or without IFN treatment. Delta-nef background yielded new hits, including IFI16. A screen in primary T cells using a reporter constructed in an alternate viral strain yielded distinct hits.

This is a clearly written paper reporting a well-performed screen for host restriction factors using replication-competent reporters, followed by exploration of a selected number of the hits. The procedure is very similar to a screen recently reported by OhAinle, as appropriately cited here.

The results are interesting and will be the subject of future study. There is not huge novelty in the approach beyond that of OhAinle, but new hits have been identified. Some further comments on overlap and lack of overlap with previous screens might be added.

While more analysis of the hits could always be added (some might want this for Nature Comm), I think the paper has achieved enough for presentation.

Small issues:

Line 44: “since more than 40 years.” – awkward

Line 288: “most likely because their genomic size is reduced by 360 bp” – I would be somewhat skeptical that this small size difference is responsible for changes in kinetics. Seems more likely that alterations in mRNA stability, splicing, or actual nef activities are responsible.

Line 315: “allowing to associate”, line 330 “allow to monitor”, and more – also awkward

Reviewer #2 (Remarks to the Author):

In this manuscript by Bozzo et al., the authors designed a new selected CRISPR screen (of 511 cellular factors, with 11 non-targeting gRNA as controls) to identify HIV restriction factors. While there are other CRISPR screens published in the field to search for HIV restriction factors (such as incorporating gRNA into HIV genome by Ohainle et al., *Elife* 2018), the major innovation of this study is the use of replication-competent HIV to identify restriction factors that are most prominent during this multi-round infection system, as opposed to previous studies using single-round HIV reporter viruses to capture one initial infection event. The strength of this manuscript is that the authors rigorously validated their results using different HIV viral strains, multiple cell lines, and in primary cells.

In Figure 1, the authors demonstrated that the positive control gRNAs targeting tetherin and GBP5 worked. The authors rigorously detected how homologous recombination looped out the insert in this replication competent system (which happens frequently) and elegantly removed the repeat regions to prevent looping out of the insert. This is rigorous and should be known to the field.

In Figure 2, the authors performed the screen with and without IFN β to test both IFN-induced and non-induced genes. The authors identified typical restriction factors IFITM1, TRIM5, and IFI16, as a sanity check and positive control that this screen works. The new finding includes identification of progranulin GRN, CIITA, EHMT2, HMXO1, and CC2D1B. The authors rigorously used two cell lines at different time points to examine the affect of these genes in HIV replication (Fig. 2b). While these genes may not always meet the significant threshold during earlier time points (such as CIITA and CC2D1B, 5 days post infection), the strength of this replication-competent HIV infection model is being able to examine the effect after multiple rounds of infection so that the effect of these genes is exponentially amplified during each round of infection (20 days post infection). EHMT2 and GRN seem to be the two most promising cellular factors having better effect size and significance.

The authors continued to validate these results individually, with and without IFN β . This is rigorous and not commonly seen in other publications in the field.

These factors are relevant to HIV infection in the literature based on:

- PGRN (GRN, progranulin), a HIV Tat binding protein (PMID 10079180) that interacts with cyclin T1.
- CIITA (MHC II transactivator) may affect T cell activation (PMID 27089879)
- EHMT2, a histone lysine methyltransferase (like EZH2), is involved in HIV silencing (PMID 28246360).
- HO-1 (encoded by HMOX1) may affect HIV replication through modulating oxidative stress (PMID 26269184).
- CC2D1B may regulate CHMP4/ESCRT-III function (PMID 22258254)

In Figure 3, the author examined whether these candidate cellular factors are IFN-induced genes (Figure 3a) and found that they are not induced by IFN – but expressed with and without IFN stimulation. Focusing on GRN (the one having the most prominent phenotype), the authors found that GRN overexpression decreases HIV LTR activity, which is slightly and partially rescued by the presence of Tat (Figure 3b). GRN is not counteracted by HIV accessory proteins (Figure 3c). GRN inhibits HIV-GFP expression in a dose dependent manner (Figure 3e). The author validated the results in primary cells (Figure 3g). Overall, Figure 3 rigorously confirms the impact of GRN on HIV infection. The authors continued to validate other cellular factors in Figure 4.

In Figure 5, the authors tested whether this sgRNA screen using a R5-tropic clinically isolated virus (as opposed to X4-tropic NL4-3) identifies different cellular factors. Some cellular factors indeed overlap, but the effect size appears different. For example, GRN and EHMT2, the two prominent hits in the NL4-3-based screen, have low effect size and significance in the CH077-based screen. RHOA, for example, is a new hit not identified in the NL4-3-based screen.

In Figure 6, the authors tested whether the absence of Nef (a viral accessory protein known to counteract many cellular restriction factors) will identify a different set of host restriction factors. The authors deleted Nef, elegantly complementing the previous Nef-competent screens. The authors found that knocking out GRN, HMOX1, CIITA and EHMT2 increase viral fitness. Serendipitously, the authors found that a viral restriction factor IFI16 (previously characterized by the authors) is counteracted by Nef.

Overall, the authors used multiple gRNA screens and identified host cellular factors that restrict HIV replication, using different viral strains, different cell lines, rigorous validation, including validations in primary cells. This is an elegant and rigorous study that should be known to the field.

Major comment:

Can the authors clarify whether Nef counteracts restriction factors identified, such as GRN, CCD2D1B, EHM2, HMOX1, and RHOA (Figure 6)?

Can the authors clarify whether these candidate restriction factors (GRN, CCD2D1B, EHM2, HMOX1, just like RHOA), affects the survival and proliferation of cells (such as primary cells or the cell lines used)? This will clarify whether the effect is because of non-specific cell death versus HIV-specific inhibition.

Does CIITA (MHC II transactivator) affects T cell activation (such as HLA-DR expression, a surrogate marker for T cell activation), and thus affects viral replication?

Minor comment:

PGRN's current name is GRN. Please make it consistent across the manuscript.

Reviewer #3 (Remarks to the Author):

Bozzo et al introduce a new method for doing CRISPR screens for finding host antiviral proteins against HIV-1 by incorporating guide RNAs into the viral genome such that viruses that inactivate an antiviral gene will have a selective advantage. The authors have identified some genes with novel antiviral properties as well as some known ones. This paper is mostly a methods description as none of the hits are followed up to any great extent. However, there are some very interesting leads that I am sure will be the subject to subsequent studies. In my mind, one of the most interesting results of the paper is the finding that a previously identified antiviral protein, IFI16, shows up most highly in the variation of the screen done with mutations in the viral nef gene, which shows the power of this

technique in identifying new targets of the HIV-1 accessory genes. Another strength of the manuscript is the use of a primary transmitter/founder virus that has a different profile of guides that are enriched.

I have only minor comments that I hope will improve the paper.

1. Line 141-142: One caveat of CRISPR screens with a custom library is that one does not find genes that are not in the library. In this case, it is not obvious to me how the genes were chosen to include. Reference 32 listed does not sound relevant, and reference 24 refers to genes under positive selection, but I did not see that the authors actually used positive selection in analyzing their hits. There are many other presumptive antiviral genes that I would have expected to come out of the screen, but did not, perhaps because they were not in the library? In any case, a clearer explanation of how the gene list was selected is warranted as well as a discussion of what which known anti-HIV genes did not come through. In that regard as well, I really like Extended Figure 2 with the positive controls to show that the system works, and I would have preferred to see that in the main text instead of the Supplement.

2. Figures 2b and 5a: If I understand correctly, what is shown in these panels are the enriched guides rather than the enriched genes. Is this correct? The MaGECK algorithm is capable of taking into account the performance of all guides to a given gene which generally gives more reliable result than using single guides. Typical CRISPR screens now use 8-10 guides per gene for this reason. Was this not possible because only 3 guides per gene were used?

3. Line 234: If I understand correctly, the gsRNA to CEACAM3 did not reduce protein levels, yet there is an effect on virus replication. Did other guides behave similarly? The most obvious explanation is that this guide targets something else and that CEACAM3 is not really the target.

4. Somewhat related to points 2 and 3, have the authors ever sequenced the entire viral genome of any of the viruses containing the “winning” guides? The reason to do so is that there is also the possibility that a gsRNA guide associated with virus that grows to higher titers has adaptive mutations elsewhere in the genome (e.g in env, or mutations in vpr that are both known to give selective advantages in long term viral growth in tissue culture). This could be one explanation of why some single guides seem to outgrow, i.e because the guide is linked to other mutations in the genome.

5. Lines 72-76: The authors have introduced a clever system with some clear advantages, but it was not really necessary for them to dismiss other CRISPR systems that have different advantages. For example, their explanation of reference 26 is incorrect. First, it is not a double transduction, but only a single transduction with a vector that has both the guide and the Cas9 followed by infections with wild-type HIV. Second, it detects guide RNAs in the viral supernatants that are enriched (not reduced) relative to the proviral copy of the guideRNA (not general viral DNA copies). Third, in contrast to the statement on line 86, this system can easily detect effects that are 2-fold even in a single round infection (as described in the original and subsequent papers) since the screen output is sequence read counts. Fourth, the system in reference 26 is more flexible than the one described in this paper since it can be used with multiple different HIV-1 strains without remaking the guide library. Fifth, the system introduced in reference 26 has been used for a comprehensive genome-wide screen for both positive and negative factors of HIV-1 infection (<https://journals.asm.org/doi/10.1128/mbio.00009-23>), to screen for differential antiviral genes on a viral mutant (<https://doi.org/10.1371/journal.ppat.1008507>), to examine HIV-1 latency factors (<https://doi.org/10.1371/journal.ppat.1011101>), and has been adapted by others to do screens directly in primary cells (<https://doi.org/10.1016/j.celrep.2023.112556>). My main point here is that the authors can more accurately describe the advantages and disadvantages of their new system and still properly acknowledge prior work.

Reply to the reviewer`s comments (in *italics*) for NCOMMS-23-59640-T

Reviewer #1 (Remarks to the Author): This paper reports a construction of a pool of HIV-1 genomes encoding sgRNAs targeting candidate restriction factors, and demonstrates the outgrowth of selected members of the library upon passage in permissive cells. (The genome needed to be engineered to prevent excision of the sgRNAs by recombination.) The benefit over earlier screens using single-cycle reporters is the ability to detect genes with weaker activity, and acting at late stages of the life cycle. (Note: it may miss genes acting very early, before the sgRNAs have acted.) New genes are identified here. The most highly enriched sgRNAs targeted six particular host genes chosen for further study. The retest of overexpression or KO of these resulted in varying effects on virus replication, with several having moderate impact on replication in cell lines and primary CD4+ cells – some effects as expected, some not. Repeats were performed with alternative viral strains and host cells, with some differences, and with or without IFN treatment. Delta-nef background yielded new hits, including IFI16. A screen in primary T cells using a reporter constructed in an alternate viral strain yielded distinct hits.

This is a clearly written paper reporting a well-performed screen for host restriction factors using replication-competent reporters, followed by exploration of a selected number of the hits. The procedure is very similar to a screen recently reported by OhAinle, as appropriately cited here.

The results are interesting and will be the subject of future study. There is not huge novelty in the approach beyond that of OhAinle, but new hits have been identified. Some further comments on overlap and lack of overlap with previous screens might be added.

While more analysis of the hits could always be added (some might want this for Nature Comm), I think the paper has achieved enough for presentation.

We thank reviewer 1 for the positive comments. For further clarification, we now explain the similarities and differences between our approach and the screen developed by OhAinle in more detail (lines 71-83). In brief, both use the CRISPR/Cas9 technology. However, in the OhAinle screen, sgRNAs and Cas9 are delivered by non-replication competent vectors harboring HIV-1 LTRs and packaging signals. In our screen, sgRNAs are expressed directly by replication-competent HIV-1, while Cas9 is expressed by the target cells. In the previous screen, two subsequent transductions/infections are required. Here, we infect Cas9 expressing T cells with sgRNA encoding fully intact HIV-1 particles in a single step. The readout in the OhAinle screen is enrichment of sgRNAs co-packaged in viral particles compared to sgRNAs in DNA isolated from cellular extracts after a single-round of virus production. We determine the enrichment and selection of specific sgRNAs in the HIV-1 population over several rounds of replication.

As appreciated by the reviewer, usage of replication-competent HIV-1 allows to enhance the impact of selective advantages at each round of replication. In addition, our approach closely reflects wild-type HIV-1 replication and antagonism of antiviral factors by accessory HIV-1 genes – with the important difference that the sgRNAs reveal their targets.

Small issues:

Line 44: “since more than 40 years.” – awkward.

Rephrased

Line 288: “most likely because their genomic size is reduced by 360 bp” – I would be somewhat skeptical that this small size difference is responsible for changes in kinetics. Seems more likely that alterations in mRNA stability, splicing, or actual nef activities are responsible.

We have previously shown that viral constructs containing deletions in nef replicate more rapidly compared to those containing premature stop codons (new Refs. 46 and 47) and didn't touch known splice sites. However, we agree that other factors may also change replication kinetics and now mention this in the revised manuscript (lines 307-301).

Line 315: “allowing to associate”, line 330 “allow to monitor”, and more – also awkward

We rephrased these statements (lines 340/341 and 355/356).

Reviewer #2 (Remarks to the Author): In this manuscript by Bozzo et al., the authors designed a new selected CRISPR screen (of 511 cellular factors, with 11 non-targeting gRNA as controls) to identify HIV restriction factors. While there are other CRISPR screens published in the field to search for HIV restriction factors (such as incorporating gRNA into HIV genome by Ohainle et al., *Elife* 2018), the major innovation of this study is the use of replication-competent HIV to identify restriction factors that are most prominent during this multi-round infection system, as opposed to previous studies using single-round HIV reporter viruses to capture one initial infection event. The strength of this manuscript is that the authors rigorously validated their results using different HIV viral strains, multiple cell lines, and in primary cells.

In Figure 1, the authors demonstrated that the positive control gRNAs targeting tetherin and GBP5 worked. The authors rigorously detected how homologous recombination looped out the insert in this replication competent system (which happens frequently) and elegantly removed the repeat regions to prevent looping out of the insert. This is rigorous and should be known to the field.

In Figure 2, the authors performed the screen with and without IFN β to test both IFN-induced and non-induced genes. The authors identified typical restriction factors IFITM1, TRIM5, and IFI16, as a sanity check and positive control that this screen works. The new finding includes identification of progranulin GRN, CIITA, EHMT2, HMXO1, and CC2D1B. The authors rigorously used two cell lines at different time points to examine the affect of these genes in HIV replication (Fig. 2b). While these genes may not always meet the significant threshold during earlier time points (such as CIITA and CC2D1B, 5 days post infection), the strength of this replication-competent HIV infection model is being able to examine the effect after multiple rounds of infection so that the effect of these genes is exponentially amplified during each round of infection (20 days post infection). EHMT2 and GRN seem to be the two most promising cellular factors having better effect size and significance.

The authors continued to validate these results individually, with and without IFN β . This is rigorous and not commonly seen in other publications in the field.

These factors are relevant to HIV infection in the literature based on:

- PGRN (GRN, progranulin), a HIV Tat binding protein (PMID 10079180) that interacts with cyclin T1.
- CIITA (MHC II transactivator) may affect T cell activation (PMID 27089879)
- EHMT2, a histone lysine methyltransferase (like EZH2), is involved in HIV silencing (PMID 28246360).
- HO-1 (encoded by HMXO1) may affect HIV replication through modulating oxidative stress (PMID 26269184).
- CC2D1B may regulate CHMP4/ESCRT-III functon (PMID 22258254)

In Figure 3, the author examined whether these candidate cellular factors are IFN-induced genes (Figure 3a) and found that they are not induced by IFN – but expressed with and without IFN stimulation. Focusing on GRN (the one having the most prominent phenotype), the authors found that GRN overexpression decreases HIV LTR activity, which is slightly and partially rescued by the presence of Tat (Figure 3b). GRN is not counteracted by HIV accessory proteins (Figure 3c). GRN inhibits HIV-GFP expression in a dose dependent manner (Figure 3e). The author validated

the results in primary cells (Figure 3g). Overall, Figure 3 rigorously confirms the impact of GRN on HIV infection. The authors continued to validated other cellular factors in Figure 4.

In Figure 5, the authors tested whether this sgRNA screen using a R5-tropic clinically isolated virus (as opposed to X4-tropic NL4-3) identifies different cellular factors. Some cellular factors indeed overlap, but the effect size appears different. For example, GRN and EHMT2, the two prominent hits in the NL4-3-based screen, have low effect size and significance in the CH077-based screen. RHOA, for example, is a new hit not identified in the NL4-3-based screen.

In Figure 6, the authors tested whether the absence of Nef (a viral accessory protein known to counteract many cellular restriction factors) will identify a different set of host restriction factors. The authors deleted Nef, elegantly complementing the previous Nef-competent screens. The authors found that knocking out GRN, HMOX1, CIITA and EHMT2 increase viral fitness. Serendipitously, the authors found that a viral restriction factor IFI16 (previously characterized by the authors) is counteracted by Nef.

Overall, the authors used multiple gRNA screens and identified host cellular factors that restrict HIV replication, using different viral strains, different cell lines, rigorous validation, including validations in primary cells. This is an elegant and rigorous study that should be known to the field.

We are pleased that reviewer 2 finds our study elegant and important.

Major comment:

Can the authors clarify whether Nef counteracts restriction factors identified, such as GRN, CCD2D1B, EHM2, HMOX1, and RHOA (Figure 6)?

In Figure 3C, we show that the accessory genes of HIV-1 have only minimal effect of susceptibility to GRN. To further address this point, we examined the effect of CCD2D1B, EHMT2, HMOX1, and RHOA overexpression on infectious WT and nef deficient HIV-1 NL4-3 and CH077 production in transfected HEK293T cells. Lack of an intact nef gene did not impact viral susceptibility to these factors (new Supplementary Fig. S9). For NL4-3 and HMOX1, we observed modest differences and will further investigate this in future studies.

Can the authors clarify whether these candidate restriction factors (GRN, CCD2D1B, EHM2, HMOX1, just like RHOA), affects the survival and proliferation of cells (such as primary cells or the cell lines used)? This will clarify whether the effect is because of non-specific cell death versus HIV-specific inhibition.

As now shown in the new Supplementary Fig. S5, depletion of these factors has no significant effect on cell viability. However, KO of RHOA slightly increased metabolic activity, which agrees with its role in cell cycling.

Does CIITA (MHC II transactivator) affects T cell activation (such as HLA-DR expression, a surrogate marker for T cell activation), and thus affects viral replication?

To address this, we examined the impact of CIITA on CD25 and HLA-DR expression. Our data shows that CIITA KO, but not GRN or control sgRNAs significantly reduced HLA-DR surface levels (new Supplementary Figure S4g). This highlights another advantage of our virus-guided approach, revealing factors supporting cellular conditions (e.g. T cell activation status) that restrict virus replication.

Minor comment:

PGRN's current name is GRN. Please make it consistent across the manuscript.

Changed throughout.

Reviewer #3 (Remarks to the Author): Bozzo et al introduce a new method for doing CRISPR screens for finding host antiviral proteins against HIV-1 by incorporating guide RNAs into the viral genome such that viruses that inactivate an antiviral gene will have a selective advantage. The authors have identified some genes with novel antiviral properties as well as some known ones. This paper is mostly a methods description as none of the hits are followed up to any great extent. However, there are some very interesting leads that I am sure will be the subject to subsequent studies. In my mind, one of the most interesting results of the paper is the finding that a previously identified antiviral protein, IFI16, shows up most highly in the variation of the screen done with mutations in the viral nef gene, which shows the power of this technique in identifying new targets of the HIV-1 accessory genes. Another strength of the manuscript is the use of a primary transmitter/founder virus that has a different profile of guides that are enriched.

I have only minor comments that I hope will improve the paper.

We appreciate the helpful comments of reviewer 3.

1. Line 141-142: One caveat of CRISPR screens with a custom library is that one does not find genes that are not in the library. In this case, it is not obvious to me how the genes were chosen to include. Reference 32 listed does not sound relevant, and reference 24 refers to genes under positive selection, but I did not see that the authors actually used positive selection in analyzing their hits. There are many other presumptive antiviral genes that I would have expected to come out of the screen, but did not, perhaps because they were not in the library? In any case, a clearer explanation of how the gene list was selected is warranted as well as a discussion of what which known anti-HIV genes did not come through. In that regard as well, I really like Extended Figure 2 with the positive controls to show that the system works, and I would have preferred to see that in the main text instead of the Supplement.

We agree that the focus on 511 cellular genes is a limitation and are currently working on establishing genome-wide libraries. All cellular targets and the corresponding sgRNAs are specified in the suppl. Table 1 and we added further information on the rationale for their selection in the revised version (lines 149-154). We are pleased that reviewer likes suppl. Fig. 2C and moved it to the main figures (new Fig. 2e)

2. Figures 2b and 5a: If I understand correctly, what is shown in these panels are the enriched guides rather than the enriched genes. Is this correct? The MaGECK algorithm is capable of taking into account the performance of all guides to a given gene which generally gives more reliable result than using single guides. Typical CRISPR screens now use 8-10 guides per gene for this reason. Was this not possible because only 3 guides per gene were used?

Generating recombinant HIV-1 constructs with high efficiency and without recombination is a challenging task and we felt that starting with targeting ~500 cellular genes each by 3 sgRNAs was an ambitious but realistic goal. However, as mentioned in the discussion section (lines 445-447) performing screens targeting all genes is useful and our aim in future studies.

3. Line 234: If I understand correctly, the gsRNA to CEACAM3 did not reduce protein levels, yet there is an effect on virus replication. Did other guides behave similarly? The most obvious explanation is that this guide targets something else and that CEACAM3 is not really the target.

The western blot analysis was most likely obscured by cross-reactivity of the antibody to closely related members of the CEACAM family of adhesion molecules. Using a new and more specific antibody, we now show in FACS analyses depletion of CEACAM3 on the protein level by ~90% (new Supplementary Fig. S4f).

4. Somewhat related to points 2 and 3, have the authors ever sequenced the entire viral genome of any of the viruses containing the “winning” guides? The reason to do so is that there is also the possibility that a gsRNA guide associated with virus that grows to higher titers has adaptive mutations elsewhere in the genome (e.g in env, or mutations in vpr that are both known to give

selective advantages in long term viral growth in tissue culture). This could be one explanation of why some single guides seem to outgrow, i.e because the guide is linked to other mutations in the genome.

Sequencing entire HIV-1 constructs expressing specific sgRNAs and associating them with mutations would technically be highly challenging and require e.g. Nanopore approaches. However, for a number of reasons, it is highly unlikely that mutations elsewhere in the viral genome are responsible for increased replication fitness: (i) the sgRNA cassettes were cloned with high efficiency and, on average, each sgRNA was covered by ~1.000 independent transformants. Thus, adaptive mutations would have to occur independently numerous times in the context of specific sgRNAs; (ii) The same sgRNAs were associated with increased replication-fitness using independently generated libraries, and independent screens in two different viral backbones. Finally, we validated selected hits in functional assays.

5. Lines 72-76: The authors have introduced a clever system with some clear advantages, but it was not really necessary for them to dismiss other CRISPR systems that have different advantages. For example, their explanation of reference 26 is incorrect. First, it is not a double transduction, but only a single transduction with a vector that has both the guide and the Cas9 followed by infections with wild-type HIV. Second, it detects guide RNAs in the viral supernatants that are enriched (not reduced) relative to the proviral copy of the guideRNA (not general viral DNA copies). Third, in contrast to the statement on line 86, this system can easily detect effects that are 2-fold even in a single round infection (as described in the original and subsequent papers) since the screen output is sequence read counts. Fourth, the system in reference 26 is more flexible than the one described in this paper since it can be used with multiple different HIV-1 strains without remaking the guide library. Fifth, the system introduced in reference 26 has been used for a comprehensive genome-wide screen for both positive and negative factors of HIV-1 infection (<https://journals.asm.org/doi/10.1128/mbio.00009-23>), to screen for differential antiviral genes on a viral mutant (<https://doi.org/10.1371/journal.ppat.1008507>), to examine HIV-1 latency factors (<https://doi.org/10.1371/journal.ppat.1011101>), and has been adapted by others to do screens directly in primary cells (<https://doi.org/10.1016/j.celrep.2023.112556>). My main point here is that the authors can more accurately describe the advantages and disadvantages of their new system and still properly acknowledge prior work.

We agree with the reviewer that the different methods have pros and cons and apologize for some inaccuracies. To address this, we introduced changes in the text (lines 71-83) and acknowledge and cite the follow-up studies using the method established by OhAinle, Emerman and colleagues.

REVIEWERS' COMMENTS

Reviewer #1 (Remarks to the Author):

The revised version has addressed the issues raised by us and the other reviewers.

The screen has identified several new host factors for future study.

Reviewer #2 (Remarks to the Author):

The authors have addressed all my questions.

Reviewer #3 (Remarks to the Author):

All of my comments have been very well addressed.